# Identification of the Genetic Basis of Response to de-Acclimation in Winter Barley

**DOI:** 10.3390/ijms22031057

**Published:** 2021-01-21

**Authors:** Magdalena Wójcik-Jagła, Agata Daszkowska-Golec, Anna Fiust, Przemysław Kopeć, Marcin Rapacz

**Affiliations:** 1Department of Plant Breeding, Physiology and Seed Science, University of Agriculture in Kraków, Podłużna 3, 30-239 Krakow, Poland; bednarczyk.an@gmail.com (A.F.); marcin.rapacz@urk.edu.pl (M.R.); 2Institute of Biology, Biotechnology and Environmental Protection, University of Silesia in Katowice, Jagiellońska 28, 40-032 Katowice, Poland; agata.daszkowska@us.edu.pl; 3The Franciszek Górski Institute of Plant Physiology, Polish Academy of Sciences, Niezapominajek 21, 30-239 Krakow, Poland; p.kopec@ifr-pan.edu.pl

**Keywords:** de-acclimation, freezing tolerance, barley, climate change, RNAseq, gene expression, oxidoreductase

## Abstract

Mechanisms involved in the de-acclimation of herbaceous plants caused by warm periods during winter are poorly understood. This study identifies the genes associated with this mechanism in winter barley. Seedlings of eight accessions (four tolerant and four susceptible to de-acclimation cultivars and advanced breeding lines) were cold acclimated for three weeks and de-acclimated at 12 °C/5 °C (day/night) for one week. We performed differential expression analysis using RNA sequencing. In addition, reverse-transcription quantitative real-time PCR and enzyme activity analyses were used to investigate changes in the expression of selected genes. The number of transcripts with accumulation level changed in opposite directions during acclimation and de-acclimation was much lower than the number of transcripts with level changed exclusively during one of these processes. The de-acclimation-susceptible accessions showed changes in the expression of a higher number of functionally diverse genes during de-acclimation. Transcripts associated with stress response, especially oxidoreductases, were the most abundant in this group. The results provide novel evidence for the distinct molecular regulation of cold acclimation and de-acclimation. Upregulation of genes controlling developmental changes, typical for spring de-acclimation, was not observed during mid-winter de-acclimation. Mid-winter de-acclimation seems to be perceived as an opportunity to regenerate after stress. Unfortunately, it is competitive to remain in the cold-acclimated state. This study shows that the response to mid-winter de-acclimation is far more expansive in de-acclimation-susceptible cultivars, suggesting that a reduced response to the rising temperature is crucial for de-acclimation tolerance.

## 1. Introduction

Under global warming, it might be considered that winter hardiness will be less critical for future crop production. However, this assumption is invalid, as the only parameters likely to change will be the predominant factors that influence the overwintering of plants locally. Climate change scenarios predict that weather conditions will become unstable, and in most cases, not typical for the season [1]. In a moderate climate zone, freezing tolerance is most important for a plant’s survival in winter. Therefore, a large body of winter hardiness-oriented research has focused on this trait. Different genes associated with freezing tolerance have been identified in many species, and the mechanisms influencing their expression have been widely studied [2,3]. In comparison, limited information is available on tolerance to de-acclimation, and the studies that have been conducted have predominantly investigated woody species [4,5].

Susceptibility to de-acclimation during winter is a complex trait. At least two types of de-acclimation with potentially distinct genetic and physiological bases can be distinguished. (1) The highest degree of freezing tolerance is attained in most plants in mid-winter. Subsequently, freezing tolerance decreases gradually. This “passive” (i.e., independent of environmental conditions) de-acclimation is connected mainly with the vegetative/reproductive transition and is widely described as the relationship between cold acclimation ability and vernalization requirements. However, it may also be associated with the decrease in organic compounds accumulated by the plant before winter and the plant’s general weakening. This type of de-acclimation is irreversible. (2) Plants also tend to de-acclimate as a result of mid-winter warm spell [1]. This “active” (in the sense of suggested reception of environmental signals) type of de-acclimation can be reversible or irreversible depending on various factors [6]. De-acclimation is unfavorable for the plant only when in spring, or after a warm period in winter, the temperature decreases rapidly to freezing temperatures [7]. Various future weather simulation models predict an increase in mean winter temperatures, which will probably cause an increase in yield loss caused by de-acclimation. Thus, tolerance to de-acclimation or ability for rapid re-acclimation will likely be critical for winter hardiness in the future [1].

Winter barley shows a relatively weak cold acclimation capability [8,9], and, in consequence, low winter hardiness, which limits large-scale production of the crop despite increasing interest from the beer industry in winter barley cultivars. The genetic basis of freezing tolerance in winter barley has been studied previously by many research groups, for example [10,11,12,13,14]. Certain genes involved in the process of cold hardening in winter barley have been identified [15,16], but the “active” de-acclimation process remains undissected.

The aim of this study was to identify genes associated with response to de-acclimation in winter barley. We assumed that mid-winter de-acclimation is not a process simply reverse to cold acclimation, and therefore, new genes associated only with active de-acclimation could be dissected.

## 2. Results

Eight previously studied (Wójcik-Jagła and Rapacz, unpublished), cold-acclimated barley accessions (four tolerant and four susceptible to de-acclimation) were subjected to de-acclimation treatment that mimicked a mid-winter warm spell (i.e., active de-acclimation). We performed differential expression analysis using RNA sequencing (RNAseq) followed by reverse-transcription quantitative real-time PCR (RT-qPCR) and enzyme activity analyses to explore the genetic basis of the response to active de-acclimation in barley.

From the differential gene expression analysis followed by comparisons using Venn diagrams, many differentially expressed genes (DEGs) were detected in various comparisons. It is emphasized that the following numbers are based on DEGs common to four accessions from each group of de-acclimation-tolerant and -susceptible genotypes.

In barley accessions tolerant to de-acclimation, 698 genes (397 upregulated and 301 downregulated) were differentially expressed between cold acclimation (CA-21) and the control (CA-0 (C)) at false discovery rate (FDR) < 0.05 and 430 genes (259 upregulated and 171 downregulated) were significant at FDR < 0.01. With regard to accessions susceptible to de-acclimation, we identified 1082 DEGs (680 upregulated and 402 downregulated) between CA-21 and CA-0 (C) with FDR < 0.05 and 747 (494 upregulated and 253 downregulated) with FDR < 0.01 (Figure 1).

Two hundred and thirteen DEGs (114 upregulated and 99 downregulated) were identified between de-acclimated (DA-28) and CA-21 for de-acclimation-tolerant accessions at FDR < 0.05, of which 115 genes (49 upregulated and 66 downregulated) were significant at FDR < 0.01. With regard to de-acclimation-susceptible accessions, 789 genes (382 upregulated and 407 downregulated) were differentially expressed in response to de-acclimation at FDR < 0.05 and 475 genes (230 upregulated and 245 downregulated) at FDR < 0.01 (Figure 2).

When compared between DA-28 and CA-0 (C) for tolerant barley accessions, 118 genes were differentially expressed (97 upregulated and 21 downregulated) at FDR < 0.05 and 57 (48 upregulated and 9 downregulated) at FDR < 0.01 (Figure 3). With respect to susceptible accessions, the same comparison identified 125 DEGs (95 upregulated and 30 downregulated) at FDR < 0.05, of which 59 (46 upregulated and 13 downregulated) were significant at FDR < 0.01 (Figure 3).

To identify genes for which expression changed owing to de-acclimation only (different from DEGs also associated with cold acclimation but regulated in the opposite direction during de-acclimation), we compared common DEGs for CA-0 (C) vs. CA-21 and DA-28 vs. CA-21. Ninety-nine DEGs (25 downregulated and 74 upregulated) were specific only to conditions mimicking de-acclimation during mid-winter warm spell in de-acclimation-tolerant barley accessions (FDR < 0.05), of which 49 genes (33 up- and 16 downregulated) were significant at FDR < 0.01 (Figure 4). In addition, 343 genes (121 downregulated and 222 upregulated) were differentially expressed explicitly after de-acclimation in susceptible barley accessions (FDR < 0.05), of which 204 genes (137 upregulated and 67 downregulated) were significant at FDR < 0.01 (Figure 4). Only 54 DEGs (19 upregulated and 35 downregulated, FDR < 0.05) specific only to de-acclimation during mid-winter warm periods were common to both de-acclimation-tolerant and -susceptible barley accessions, of which seven genes (five upregulated and two downregulated) were significant at FDR < 0.01 (Figure 5).

Gene ontology (GO) enrichment analysis revealed significant GO terms for DEGs specific only to de-acclimation (Table 1 and Appendix A). The susceptible group of accessions showed a much more functionally diverse spectrum of genes for which expression was changed in response to temperature rise. A considerable number of DEGs from both tolerant and susceptible accessions was directly or indirectly associated with photosynthesis. Thirty-nine sequences were associated only with phosphorylation (GO:0016310), and 27 were associated with phosphate-containing compound metabolic processes (GO:0006796), eight with ion transmembrane transport (GO:0034220), and 41 were significantly associated with either ATP binding (GO:0005524) or ATP metabolic processes (GO:0046034; Table 1).

A total of 224 sequences of DEGs specific to de-acclimation (FDR < 0.01) were successfully annotated with either characterized or uncharacterized proteins. In addition, the most similar sequences were identified using the BLAST algorithm. Thirty-six sequences were annotated only in de-acclimation-tolerant barley accessions, whereas 181 sequences were characteristic solely of de-acclimation-susceptible accessions. Seven annotated sequences were common to de-acclimation-susceptible and -tolerant accessions (Table 2).

Five annotated DEGs were selected for further verification in an RT-qPCR experiment. The selected sequences were annotated with characterized proteins. The average (for four accessions) log_2_ fold change in their expression between CA-21 and DA-28 was at least 4.0 or −4.0. In addition, the function of the annotated protein was a crucial consideration in the selection process. The chosen sequences were annotated to proteins belonging to four groups: Stress response—antioxidative enzymes (peroxidase and catalase), stress response—heat shock proteins (sHSP domain-containing protein), stress response—freezing tolerance-related proteins (CBF 14), and proteins involved in structural functions of cell walls and membranes (LRRNT_2 domain-containing protein/polygalacturonase (PGU) inhibitor-like).

The transcription profiles of *Peroxidase* did not show a consistent pattern among the studied de-acclimation-tolerant and -susceptible barley accessions (Figure 6). Increase in transcript copy number after de-acclimation (DA-28) was observed in the accessions Aday-4, Astartis, Aydanhanim, Carola, and Mellori, of which the latter four were classified as de-acclimation-susceptible in a previous study (data not published). The increase in *Peroxidase* transcript accumulation was preceded by an initial decrease at DA-23 in most cases (Figure 6). The expression profiles of *Catalase* were almost identical for half of the tested barley accessions, namely, Aday-4, DS1022, Pamina (de-acclimation tolerant), and Carola (de-acclimation susceptible) (Figure 6). In these accessions, a slight increase in *Catalase* expression was observed at the beginning of de-acclimation, and a distinct increase was detected after one week under de-acclimating conditions followed by a dramatic decrease in transcript copy number during re-acclimation to cold. An increase in expression of *Catalase* at DA-28 in relation to CA-21 was also observed in DS1028, Astartis (de-acclimation tolerant), and Aydanhanim (de-acclimation susceptible). In these accessions, a more significant (or equally high) number of copies of the *Catalase* gene was detected at DA-23 (Figure 6).

Accumulation of *sHSP* transcripts was distinctly higher at DA-23 and DA-28 in relation to that of CA-21 in all tested barley accessions, regardless of their tolerance to de-acclimation (Figure 6). However, expression drastically decreased after one week of re-acclimation in all accessions. Three types of expression patterns were distinguishable for *sHSP*: The same level of *sHSP* transcripts at the DA-23 and DA-28 time points (Aday-4, Astartis, and Mellori), an abrupt increase in expression at the beginning of de-acclimation followed by a slight decrease after seven days of de-acclimation (Pamina, Carola, and DS1022), and a gradual increase in *sHSP* transcript accumulation from the beginning of de-acclimation and peaking after seven days of de-acclimation (Aydanhanim and DS1028) (Figure 6). The expression of *cbf14* did not change or slightly decreased at the DA-23 and DA-28 time points in relation to CA-21 in all tested barley accessions (Figure 6).

Higher accumulation of *PGU inhibitor-like* transcripts during and after de-acclimation in relation to CA-21 was observed in all tested barley accessions except Mellori (Figure 6). In Mellori, the transcript level did not change in response to de-acclimation. Three patterns of expression of the PGU inhibitor-like protein-coding gene were observed among the remaining seven accessions: A significant increase in transcript level at DA-23 with the level maintained after seven days of de-acclimation (Aday-4, Astartis, and DS1028), a gradual increase in transcript level starting from DA-23 with the peak at DA-28 (Pamina, Carola, and DS1022), and a significant increase in transcript level at DA-23 with reduced accumulation of transcripts observed after completion of de-acclimation (Aydanhanim) (Figure 6). 

An apparent increase in ascorbate peroxidase activity after de-acclimation (DA-28) compared with that under cold acclimation (CA-21) was observed in five (Aday-4, DS1022, Pamina, Astartis, and Mellori) of the eight tested barley accessions (Figure 7). In four of the former accessions, ascorbate peroxidase activity decreased or remained unchanged at the beginning of de-acclimation (DA-23). In Astartis ascorbate peroxidase activity had already started to increase at DA-23. No changes in the activity of this enzyme owing to de-acclimation were observed in DS1028. In Aydanhanim the activity rose at DA-23, but drastically decreased after seven days of de-acclimation (DA-28). The pattern of changes in ascorbate peroxidase activity caused by de-acclimation in Carola was the opposite to that observed in Aydanhanim –activity decreased significantly at DA-23 and at DA-28 returned to a level similar to that recorded at CA-21 (Figure 7). 

An increase in glutathione peroxidase activity after de-acclimation (DA-28) in relation to that of cold-acclimated plants (CA-21) was observed in three tested barley accessions—DS1022, DS1028, and Pamina—which were all classified as tolerant to de-acclimation in previous experiments (data not published) (Figure 7). In Pamina, this increase in activity was most distinct and was preceded by a decrease in activity at the beginning of de-acclimation (DA-23). In Astartis, the glutathione peroxidase activity decreased initially during de-acclimation but returned to the CA-21 level after seven days of de-acclimation. In Mellori, a slight initial increase in activity was observed at DA-23, followed by a decrease leading to the same level of activity recorded at CA-21. In Aydanhanim, Aday-4, and Carola, glutathione peroxidase activity decreased during and after de-acclimation compared with cold-acclimated plants. The decrease was most drastic in Aydanhanim (Figure 7). 

Changes in guaiacol peroxidase activity caused by de-acclimation showed different patterns among the barley accessions (Figure 7). In Aday-4, DS1028, and Carola, activity was lower during and after de-acclimation compared with that recorded for cold-acclimated plants. In DS1028 and Carola, activity rose at DA-28 compared with that at DA-23, but did not attain the level of activity observed after cold acclimation (CA-21). In Astartis and Mellori, a slight decrease in guaiacol peroxidase activity was observed at the beginning of de-acclimation but was followed by a considerable increase after one week of de-acclimation, attaining higher activity than that observed in cold-acclimated plants. In Aydanhanim, DS1022, and Pamina, the guaiacol peroxidase activity was higher during (DA-23) and after (DA-28) de-acclimation than after cold acclimation (CA-21). In DS1022 and Pamina, the activities recorded at the DA-23 and DA-28 time points were similar, whereas in Aydanhanim, the guaiacol peroxidase activity at DA-28 was distinctly lower than that at DA-23 (Figure 7). 

The pattern of nonspecific peroxidase activity differed among all of the tested barley accessions, but some similarities were observed (Figure 7). The activity increased initially during de-acclimation in DS1028 and Pamina, then decreased to a level similar to that recorded for cold-acclimated plants after seven days of de-acclimation. The profile of changes caused by de-acclimation was similar for Aydanhanim, but the decrease at DA-28 was smaller, but the activity remained higher at DA-28 than in CA-21. In Mellori nonspecific peroxidase activity gradually increased owing to de-acclimation and decreased rapidly during re-acclimation to cold. In Carola and DS1022, the initial decrease in nonspecific peroxidase activity observed at DA-23 was followed by a rapid increase at DA-28, resulting in higher activity than that recorded in CA-21. In Aday-4 a decrease in nonspecific peroxidase activity during and after de-acclimation was observed. No changes in nonspecific peroxidase activity caused by de-acclimation were observed for Astartis (Figure 7). 

The profile of changes in formate dehydrogenase activity caused by de-acclimation was similar for five barley accessions (Figure 8). In Astartis, Aydanhanim, Carola, DS1028, and Pamina, activity increased considerably in the initial stage of de-acclimation (DA-23) and decreased rapidly after seven days of de-acclimation. The decrease led to activity lower than that observed in CA-21 in four of the accessions. In Aday-4 and Mellori, the formate dehydrogenase activity was lower during and after de-acclimation compared with that of cold-acclimated plants. The activity remained low also during re-acclimation to cold. In DS1022, formate dehydrogenase activity increased during and after de-acclimation, and the trend towards increase continued during re-acclimation (Figure 8). 

The typical pattern of change in NADPH cytochrome P450 reductase activity was a significant increase in response to cold acclimation (CA-21) in all tested barley accessions (Figure 8). In some accessions (Carola, Mellori, and Pamina), the increase was notable at the beginning of cold acclimation (CA-7). In DS1028, the activity remained high at the beginning of de-acclimation and decreased rapidly by the end of de-acclimation treatment. In the remaining accessions, NADPH cytochrome P450 reductase activity decreased abruptly in the initial stage of de-acclimation (DA-23). A slight increase in activity by the end of de-acclimation was observed in Carola and DS1022, and this trend continued during re-acclimation to cold (Figure 8). 

Four accessions, namely, Aydanhanim, Carola, DS1022, and Pamina, displayed an increase in catalase activity induced by de-acclimation (DA-23) followed by a substantial decrease after one week of de-acclimation (DA-28; Figure 8). This pattern was much more pronounced in Aydanhanim, DS1022, and Pamina than in Carola. Astartis also showed an increase in catalase activity caused by de-acclimation, but only by the end of the treatment (DA-28). Mellori was the only cultivar to show no response in catalase activity to de-acclimation. Aday-4 and DS1028 showed a steady decrease in catalase caused activity by de-acclimation treatment (Figure 8).

## 3. Discussion

Limited information is available on the molecular control of the response to de-acclimation in herbaceous plants. To the best of our knowledge, only one previous study has examined control at the DNA level using genome-wide association mapping [17], and that study was performed on a dicotyledonous species. Furthermore, few proteomic studies have explored changes associated with de-acclimation [18,19]. The majority of transcriptomic analyses, which represent the most common molecular investigations of de-acclimation, have used *Arabidopsis thaliana* as the experimental material [20,21,22,23,24]. Arabidopsis is a model plant with limited relevance to cereals. The conditions used for cold acclimation and de-acclimation in previous studies are not entirely relevant to the field conditions under which cereals are grown. Studies of other plant species, including grasses, also have employed a broad range of approaches to de-acclimation treatments [6,25,26,27,28,29,30,31,32,33,34,35]. De-acclimation conditions applied in previous studies often more closely resemble spring warming than mid-winter warm spell, using equal night and day lengths or longer days/shorter nights sometimes accompanied by relatively high temperatures [6,25,28,35]. Moreover, most of these studies describe physiological and biochemical changes caused by de-acclimation in herbaceous plants, but not their molecular background.

In the only previous study of the molecular background of changes caused by de-acclimation in barley available to date, microRNAs isolated during de-acclimation were identified [28]. In that study, the most significant number of differentially expressed microRNAs was observed on the sixth day of cold de-acclimation, which corresponds to seven days of de-acclimation in the present RNAseq analysis. Although drawing deductions from the applied de-acclimation treatment, the previous study is actually concerned with spring-type de-acclimation events, and one of the two barley cultivars studied is a spring cultivar. Nevertheless, some of the results reported are similar to those obtained in the current study. MicroRNAs targeting two peroxidases and 15 other oxidoreductases were detected in the winter barley cultivar Nure [28], which is in agreement with four peroxidase- and seven other oxidoreductase-coding transcripts identified in the present study. In addition, C-repeat binding factor (CBF), late embryogenesis abundant (LEA), and auxin response protein-encoding genes identified in the present study were previously recognized as targeted by microRNAs that were differentially expressed in response to de-acclimation [28].

In the current study, a large number of transcripts were associated with the response to de-acclimation both in de-acclimation-tolerant and -susceptible barley accessions under FDR < 0.05. A considerable number of these transcripts remained significant at FDR < 0.01. In contrast to some studies [18], in which most of the changes detected during de-acclimation were simply opposite to changes observed during acclimation to cold, the present study showed that a higher number of transcripts are characteristic only to de-acclimation or only to cold acclimation than are common to both responses (but are regulated in the opposite direction) (Figure 4). Furthermore, we showed that most DEGs associated specifically with de-acclimation in barley differs between de-acclimation-susceptible and -tolerant accessions (Figure 5). The DEG analysis also revealed a substantially higher number of de-acclimation-induced expression changes in de-acclimation-susceptible accessions than in de-acclimation-tolerant cultivars (Figure 5). These findings may indicate that the deciding factor determining the survival of frost events after a mid-winter warm period is not mechanisms that confer tolerance, but rather an insensitivity to temperature rise, which triggers a set of metabolic or developmental changes in de-acclimation-susceptible accessions associated with up- and downregulation of genes. The differences in the number of DEGs associated with de-acclimation and the scarcity of common transcripts suggest that de-acclimation-tolerant and-susceptible genotypes exhibit distinct genetic responses to mid-winter active de-acclimation, which could select de-acclimation-tolerant genotypes. 

Curiously, in the present study, no significant GO enrichment terms in the “cell component” category were identified. Previous reports focusing on a cell component, namely, the plasma membrane, have examined aspects of plant de-acclimation [18]. The present GO analysis identified photosynthesis-related molecular functions and biological processes as the most highly enriched categories. The role of photosynthesis in response to de-acclimation has been discussed previously by several authors [6,21,22,30], and transcripts of photosynthesis-related genes have been identified in several transcriptomic studies [20,22,23,36]. However, the present GO enrichment analysis did not entirely correspond to the results of a more thorough bioinformatic analysis leading to the annotation of specific genes (Table 2). 

Although the gene annotation performed in the present study also revealed a large number of DEGs involved in photosynthesis or associated with chloroplasts, a distinct overrepresentation of genes encoding oxidoreductases, especially peroxidases, was noted, which the GO analysis did not reveal. It was previously suggested that redox enzymes might play a crucial role in the de-acclimation response [4,27], thus associating the susceptibility to freezing after a warm period, with reduced tolerance to oxidative stress. This suggestion would imply the downregulation rather than upregulation of genes encoding antioxidant enzymes, but the results from the present DEG analysis revealed the upregulation of all genes from this group (Table 2). Similar results were obtained previously [21,22,28]. In the present study, the significant de-acclimation-related DEGs encoding oxidoreductases were only identified among the de-acclimation-susceptible barley cultivars. Given that these DEGs were also overrepresented compared with other annotated genes, the results indicate that susceptibility to mid-winter de-acclimation in barley is predominantly caused by overly rapid activation of defense mechanisms against reactive oxygen species. Perhaps excessively early mobilization of oxidative stress defense mechanisms results in the plant not responding sufficiently quickly to the real threat of freezing temperatures. This hypothesis is supported by most RT-qPCR and oxidoreductase activity results observed at the RA-35 (re-acclimation) time point in this study, wherein the transcript abundance and enzyme activity decreased rapidly after peaking during de-acclimation treatment (Figure 6 and Figure 7).

However, it was previously suggested that the upregulation of gene expression is followed by post-transcriptional suppression of antioxidants during de-acclimation, leading to reduction of stress tolerance in barley [28]. As already mentioned, a larger majority of DEGs associated with de-acclimation was detected in the susceptible group of cultivars compared with the de-acclimation-tolerant cultivars, and most of those DEGs were upregulated. This result suggests the superiority of genotypes that do not show a response to de-acclimation. The genotypes that de-acclimate may initiate post-stress recovery mechanisms, hence the upregulation of antioxidant enzyme-coding genes, but also other stress-related and cytoskeleton-related genes identified in the current study. The GO enrichment analysis also revealed upregulation of genes associated with ATP binding and protein phosphorylation (Table 1, Appendix A), which suggests these genes participate in post-transcriptional protein modifications. The changes may relate to protein reprogramming owing to the temperature rise [37,38,39]. This response also emphasizes the difference in perception of the two basic types of de-acclimation—active (mid-winter) and passive (spring). Spring warming events also involve recovery mechanisms, but predominantly trigger signals for plant development [4], and almost no plant development-related genes were annotated in the present study (Table 2). Furthermore, downregulation of genes involved in the control of ATP synthesis and energy coupled proton transport (Table 1, Appendix A) might reflect decreased energy demand, which normally increases during acclimation to cold. A lowered requirement for energy also suggests that no rapid growth/developmental processes are triggered in actively de-acclimated barley plants. The reason may relate to day length, which is markedly shorter in mid-winter than in early spring, and night temperatures, which are lower in winter compared with those in early spring.

Among annotated transcripts revealed to be associated with the de-acclimation response in the current study were LEA-coding genes (Table 2). Previous studies have noted an association of LEA proteins with de-acclimation [25,28,31]. Identification of auxin response protein-coding genes among the DEGs upregulated in de-acclimation-tolerant barley accessions in the present study is consistent with previous reports [20,22,23,28]. Many genes associated with stress response in plants were identified in the present study, including oxidoreductase-coding genes, heat shock protein-coding genes, pathogen response-associated genes (of which the core response is similar to the freezing stress response), and freezing stress-related genes, namely, CBFs. Genes belonging to all of these groups, especially *CBF* genes, were previously reported as associated with de-acclimation in herbaceous plants [20,22,23,24,28].

The results of the present RT-qPCR experiments confirmed the changes in expression of the selected genes associated with the response to de-acclimation in the majority of cases (Figure 6). No expected changes related to mid-winter de-acclimation were observed only in the *cbf14* expression profile (Figure 6). This result may be associated with the specific time-dependent character of *cbf* gene expression, which usually peaks within the first 12–24 h of stress treatment [40]. There is a possibility that the timing of collection of samples for the RNAseq and RT-qPCR experiments differed sufficiently to affect the detection of their expression despite our careful efforts to repeat the experimental conditions. For the remainder of the selected genes, namely, peroxidase, catalase, sHSP, and PGU inhibitor-like coding genes, upregulation during and after seven days of de-acclimation was observed in most of the barley accessions irrespective of their tolerance to mid-winter de-acclimation (Figure 6). These results may partly reflect that the comparisons made for detecting differential transcripts using Venn diagrams [41] showed only DEGs common for all of the four de-acclimation-tolerant or four susceptible barley accessions. In addition, certain DEGs could also be expressed in some members of the other group. That was, indeed, the case for all of the RT-qPCR-tested genes where the gene identified as differentially expressed in response to de-acclimation in all of the four susceptible genotypes was also differentially expressed in one (*cbf14*), two (*Peroxidase*, *Catalase*, and *sHSP*), or three (*PGU inhibitor-like*) tolerant accessions (data not shown). 

The overrepresentation of different types of oxidoreductase gene transcripts among the DEGs responsive to de-acclimation in barley showed the necessity for an enzyme activity analysis of certain selected oxidoreductases, mostly peroxidases, under the same conditions as those applied for the RT-qPCR experiment. The changes observed in the activity of the selected enzymes did not correspond or corresponded only partially to the changes in the number of accumulated transcripts of genes encoding peroxidases and catalase (Figure 7 and Figure 8). As described previously [42], the number of accumulated transcripts might not comply with the amount of accumulated protein for various reasons. Additional proteomic analysis would be helpful in the future to provide a comprehensive overview of the role of oxidoreductase enzymes in response to mid-winter de-acclimation in barley.

In conclusion, although certain portions of the response to mid-winter warm spell-induced de-acclimation are the reverse of the response to cold acclimation, the molecular backgrounds of these two processes’ predominantly differ. The present study provides novel evidence for the distinct molecular regulation of cold acclimation and de-acclimation. In addition, mid-winter active de-acclimation is regulated differently from that of passive spring de-acclimation, which is associated with developmental changes. De-acclimation in mid-winter is indicated to be perceived as an opportunity to regenerate after stress. Unfortunately, it is competitive to remain in the cold-acclimated state, which can be deduced from the majority of genes for which expression is activated under de-acclimation. Antioxidant enzymes and other oxidoreductases seem to play a crucial role in the process of active de-acclimation, but there is still insufficient evidence to link their abundance with the degree of barley tolerance to de-acclimation. Photosynthesis-related processes may be of fundamental importance during de-acclimation, as deduced from GO enrichment analysis, but unambiguous confirmation is required. Nonetheless, the present study demonstrates that the response to mid-winter de-acclimation is far more expansive in de-acclimation-susceptible cultivars, suggesting that the key to de-acclimation tolerance is a passive or muted response to the rise in temperature.

## 4. Materials and Methods 

### 4.1. Plant Material and Growth Conditions

Four winter barley lines and cultivars tolerant to de-acclimation (Aday-4, DS1022, DS1028, and Pamina) and four de-acclimation-susceptible accessions (Aydanhanim, Astartis, Carola, and Mellori) selected previously (Wójcik-Jagła and Rapacz, unpublished) were used in this study. Seeds were sown in plastic pots (5 dm^3^, one genotype per pot and one pot per genotype, 12 seeds per genotype) filled with a mixture of universal garden soil substrate (Ekoziem, Jurkow, Poland) and sand (1:1, v/v). The pots were transferred to a growth chamber after sowing (darkness, 25 °C/17 °C [day/night]). Irradiance of 400 μmol m^−2^ s^−1^ (HPS lamps, SON-T+ AGRO, Philips, Brussels, Belgium) under a photoperiod of 12 h/12 h (light/dark) was provided when the seedlings started to emerge. The temperature was reduced to 15 °C/12 °C (day/night) 8 days after sowing. The plants were subjected to 3 weeks cold-hardening 20 days after sowing (4 °C/2 °C [day/night], photoperiod of 9 h/15 h [light/dark], and irradiance of 250 μmol m^−2^ s^−1^). After 3 weeks acclimation to cold, the plants were subjected to de-acclimation (7 days of 12 °C/5 °C [day/night]).

### 4.2. RNA Isolation

Leaves from each genotype were sampled before (CA-0 (C)) and after cold acclimation (CA-21), and after de-acclimation (DA-28) in three biological replicates (leaves from three different plants). Samples were immediately frozen in liquid nitrogen and stored at −80 °C until use. Total RNA was isolated from 72 leaf samples (0.03–0.05 g from the middle portion of the youngest fully developed leaf) using the RNeasy Plant Mini Kit (Qiagen, Hilden, Germany). The quantity and purity of RNA was checked using a UV-Vis Q5009 spectrophotometer (Quawell, San Jose, CA, USA). RNA integrity was tested using a 2100 Bioanalyzer (Agilent Technologies, Santa Clara, CA, USA).

### 4.3. RNA Sequencing and Differential Expression Analysis

Total RNA from the 72 samples was submitted to Genomed (Warsaw, Poland) for sequencing. The RNA was subjected to mRNA isolation using the NEBNext Poly(A) mRNA Magnetic Isolation Module (New England Biolabs Inc., Ipswich, MA, USA). The libraries were prepared with the NEBNext Ultra Directional RNA Library Prep Kit for Illumina (New England Biolabs) and sequenced using a HiSeq 4000 platform (Illumina, San Diego, CA, USA) in PE101 mode. Received raw sequence data were subjected to FastQC analysis to check the quality of reads and presence/absence of adapters [43]. The BAC-based barley reference sequence [44] was used to map the RNA-seq data. Read count and transcripts per million reads mapped data were determined using Kallisto version 0.43.0 software [45]. Differential expression analysis was performed using DeSeq2 [46] to compare the transcriptomes of control (pre-hardening), cold-acclimated, and de-acclimated plants. The FDR was primarily set as <0.05 so as not to overlook interesting but weakly significant interactions, and then reduced to <0.01 to simplify the selection of genes for further verification via RT-qPCR.

Obtained data sets were grouped and contrasted using Venn diagrams [41]. Comparisons were made for control vs. cold-acclimated (CA-0 (C)/CA-21), cold-acclimated vs. de-acclimated (CA-21/DA-28), and de-acclimated vs. control (DA-28/CA-0 (C)) for de-acclimation-tolerant and -susceptible accessions separately and also for common DEGs. The DEGs were then subjected to GO analysis using the AgriGo online toolkit with singular enrichment analysis [47] using the default settings (FDR < 0.05). 

The Horvu sequences were annotated to specific proteins using the Uniprot database [48] and aligned to determine similarities with closely related species using the NCBI Blast tool [49]. 

### 4.4. Gene Expression Analysis

Five genes were selected for verification of their expression under de-acclimation treatment. The genes were selected on the basis of GO analysis, annotation, and the magnitude of expression changes in response to de-acclimation revealed by differential expression analysis. Primer and probe sequences (Table 3) were designed for these genes using Primer3Plus [50,51] based on consensus sequences (when more than one splicing variant was possible) derived from the EnsemblPlants.org database [52,53]. For the alignment of two splicing variants, the pairwise alignment tool Lalign [54] was used. In comparison, the multiple alignment tool Clustal Omega [55], as well as Kalign [56] were used for aligning three or more variants.

RNA for gene expression analysis was isolated from leaves of the genotypes used for RNAseq using the aforementioned method. The growth, cold acclimation, and de-acclimation conditions were the same as described in Section 4.1, but the plants were also subjected to re-acclimation (same conditions as for cold acclimation but treated for 10 days). Leaves were sampled with three biological replications (leaves from three individual plants) at five time points: CA-7, during cold acclimation (1 week after moving the plants to the hardening conditions); CA-21, after cold acclimation; DA-23, during de-acclimation (2 days after moving the plants to the de-acclimating conditions), DA-28, after de-acclimation; and RA-35, during re-acclimation to cold (after seven days). To receive template cDNA the RNA was subjected to reverse transcription using the QuantiTect Reverse Transcription Kit (Qiagen, Hilden, Germany) reagent set. We used RT-qPCR analysis to determine changes in expression of the selected genes. The reactions were performed using a QuantStudio 3 Real-Time PCR System (Thermo Fisher Scientific, Waltham, MA, USA). Amplification was observed from the increase in fluorescence intensity of SYBRGreen (for reference genes [57]) and 6-carboxyfluorescein (FAM) from TaqMan MGB probes (for analyzed genes [58,59]). The reference genes were the ADP-ribosylation factor 1-like protein (ADP) and *S*-adenosylmethionine decarboxylase (sAMD) coding genes [11]. The reactions were conducted in three biological replicates for each genotype, each in three instrumental replicates. Each reaction contained 900 nM of each primer, approximately 35 ng cDNA template, and TaqMan™ Gene Expression Master Mix (Thermo Fisher Scientific, Waltham, MA, USA).

The relative level of expression of the analyzed genes was calculated using the modified standard curve method [60]. The expression level during cold hardening (CA-7) normalized in relation to the geometric mean of the internal standard genes’ copy number was used as a reference time point (number of gene copies = 1) for all of the other tested time points. The standard error was calculated for the geometric means of three instrumental replications × three biological replications × two reference genes.

### 4.5. Analysis of Oxidoreductase Activity

The samples for analysis of oxidoreductase activity were collected at the same time points as for the gene expression analysis plus an additional control time point, CA-0 (C), before cold acclimation. One sample consisted of one fully developed leaf, either the first, second, or third leaf, depending on the developmental stage attained at a particular time point. The weight of the leaves ranged from 0.12 to 0.59 g. Each line at each time point was represented by three biological replications (three leaves from three individual plants). The standard error was calculated for the mean of three repetitions at each time point. The activity of seven enzymes was measured: Ascorbate, glutathione, guaiacol, and nonspecific peroxidases, as well as catalase, formate dehydrogenase, and NADPH-cytochrome P450 reductase.

Leaves were homogenized in 50 mM Tris–HCl buffer (pH 7.8) supplemented with 1 mM EDTA-Na_2_, 3% polyvinylpyrrolidone, and 1% Protease Inhibitore Coctail (Merck, Darmstadt, Germany). Homogenization buffer was added in the proportion of 6 µL buffer per 1 mg of plant material. The homogenate was centrifuged at 12,000 *g* for 20 min at 4 °C. The supernatant was used for further analysis. Enzyme activity was measured spectrophotometrically using a Synergy 2 Microplate Reader (BioTek, Winooski, VT, USA). The activity of the enzymes was normalized to the amount of protein. Protein content was determined in accordance with Reference [61], using bovine serum albumin as a standard. 

Ascorbate peroxidase activity was determined as the oxidation of ascorbic acid, in accordance with Reference [62]. The reaction mixture (114 µL) consisted of 100 µL buffer (50 mM Tris–HCl buffer, pH 7.8), 5 µL of 590 mM ascorbic acid, 5 µL of 19.6 mM H_2_O_2_, and 4 µL leaf extract. Enzyme activity was measured by monitoring the decline in absorbance at 290 nm (extinction coefficient of ascorbic acid was 2.8 mM^−1^ cm^−1^). The respective control reaction mixtures contained buffer instead of H_2_O_2_ solution. The ascorbate peroxidase activity was expressed as nmol ascorbate mg^−1^ protein min^−1^. 

Glutathione peroxidase activity was assayed by measuring the decrease in NADPH concentration, in accordance with Reference [63]. The assay mixture (126 µL) contained 100 µL buffer (50 mM Tris–HCl, pH 7.0, 0.5 mM EDTA-Na_2_, and 1.6 mM NaN_3_), 5 µL of 19.52 mM reduced glutathione, 5 µL of 2.69 mM NADPH, 5 µL of 3 U mL^−1^ glutathione reductase, 5 µL of 4.5 mM H_2_O_2_, and 6 µL leaf extract. The absorbance was monitored at 340 nm (extinction coefficient of NADPH was 6.2 mM^−1^ cm^−1^). The control reaction was performed using the buffer instead of H_2_O_2_ solution. The glutathione peroxidase activity was expressed as nmol NADPH mg^−1^ protein min^−1^. 

The activity of nonspecific peroxidases was measured using two methods exploiting different substrates. Nonspecific peroxidase 1 (Guaiacol peroxidase) activity was determined by measuring the formation of the conjugate of guaiacol, in accordance with Reference [64]. The reaction medium (98 µL) contained 88 µL buffer (0.1 M sodium phosphate buffer, pH 6.5), 4 µL of 2 mM guaiacol, 4 µL of 0.13 M H_2_O_2_, and 2 µL leaf extract. The kinetic evolution was measured at 436 nm (extinction coefficient of ascorbic acid was 26.6 mM^−1^ cm^−1^ for the conjugate). The guaiacol peroxidase activity was expressed as µmol tetraguaiacol mg^−1^ protein min^−1^. Nonspecific peroxidase 2 activity was measured using *p*-phenylenediamine (pPD) as the substrate, in accordance with Reference [65]. The reaction mixture (249.5 µL) contained 236.5 µL of buffer (0.3 M potassium phosphate buffer, pH 7.8, and 0.1 mM EDTA-Na_2_), 1.5 µL of 0.5% pPD, 10 µL of 9.8 M H_2_O_2_, and 1.5 µL leaf extract. The nonspecific peroxidase 2 activity was calculated as the change in absorbance at 485 nm (∆A) per minute and normalized to micrograms of protein. 

Catalase activity was measured based on the rate of H_2_O_2_ decomposition at 240 nm (extinction coefficient of H_2_O_2_ was 0.04 mM^−1^ cm^−1^), in accordance with Reference [66]. The assay mixture (107.5 µL) was composed of 100 µL buffer (0.1 M sodium phosphate buffer, pH 6.5), 5 µL of 134.5 mM H_2_O_2_, and 2.5 µL leaf extract. The catalase activity was expressed as mmol H_2_O_2_ mg^−1^ protein min^−1^.

NADPH-cytochrome P450 reductase activity was determined by measuring the oxidation of cytochrome *c* (Cyt. c), in accordance with Reference [67]. The assay medium (200 µL) consisted of 166 µL buffer (0.3 M potassium phosphate buffer, pH 7.8, and 0.1 mM EDTA-Na_2_), 20 µL of 10 mM NADPH, 16 µL of 0.5 mM cytochrome *c*, and 16 µL leaf extract. The absorbance was monitored at 550 nm (extinction coefficient of the oxidized form of Cyt. c was 21 mM^−1^ cm^−1^). The NADPH-cytochrome P450 reductase activity was expressed as nmol Cyt. cox mg^−1^ protein min^−1^.

Formate dehydrogenase activity was determined by measuring the production of NADH, in accordance with Reference [68]. The reaction mixture (250 µL) contained 195 µL buffer (0.1 M sodium phosphate buffer, pH 6.5), 25 µL of 0.5 M sodium formate, 25 µL of 10 mM NAD+, and 5 µL leaf extract. The kinetic evolution was measured at 550 nm after incubation at 30 °C (extinction coefficient of NADH was 6.2 mM^−1^ cm^−1^). The formate dehydrogenase activity was expressed as nmol NADH mg^−1^ protein min^−1^.

## Figures and Tables

**Figure 1 ijms-22-01057-f001:**
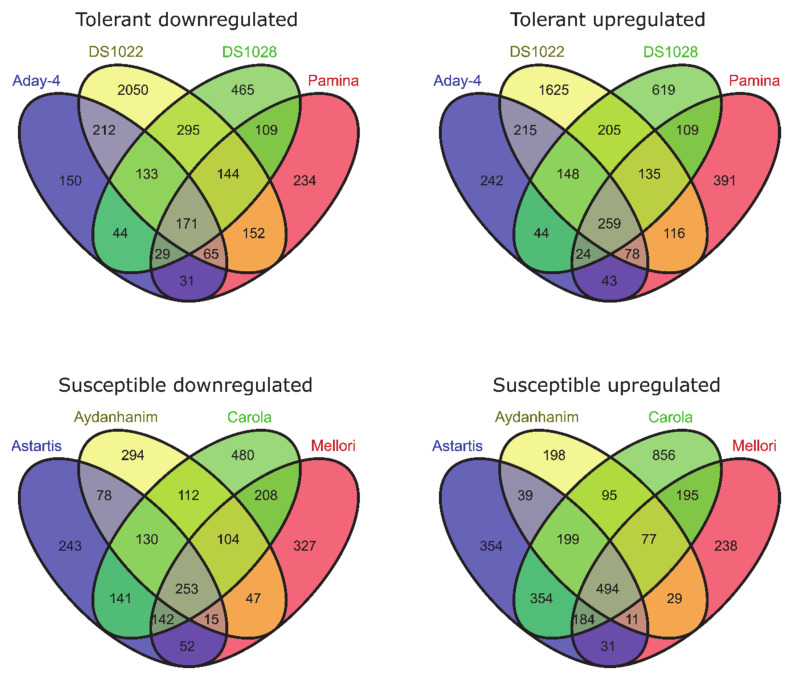
Differentially expressed genes (DEGs) between cold-acclimated (CA-21) and control (CA-0 (C), before cold acclimation) barley accessions (log_2_FC = 2, false discovery rate (FDR) < 0.01).

**Figure 2 ijms-22-01057-f002:**
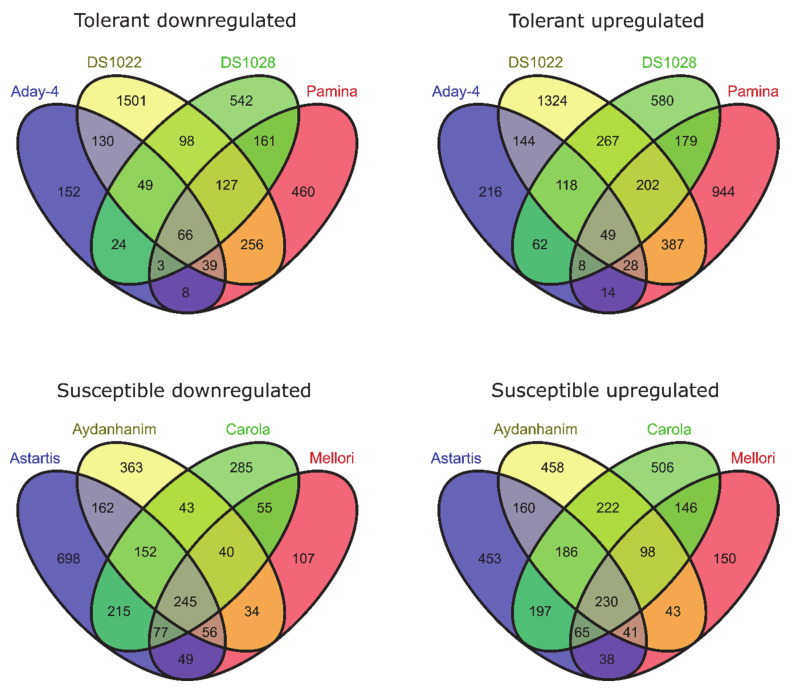
DEGs between de-acclimated (DA-28) and cold-acclimated (CA-21) barley accessions (log_2_FC = 2, FDR < 0.01).

**Figure 3 ijms-22-01057-f003:**
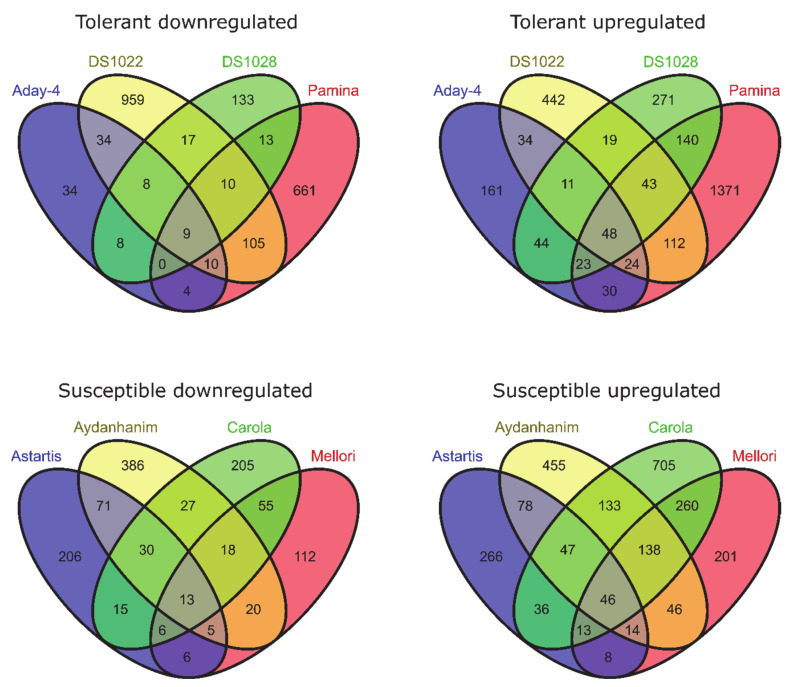
DEGs between de-acclimated (DA-28) and control (C0, before cold-acclimation) barley accessions (log_2_FC = 2, FDR < 0.01).

**Figure 4 ijms-22-01057-f004:**
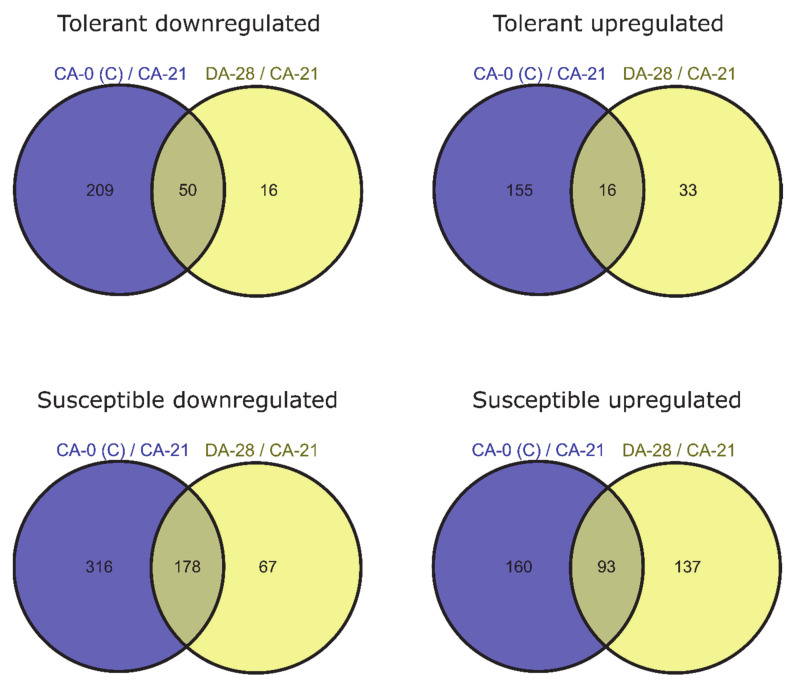
DEGs specific only to de-acclimation (yellow) during mid-winter warm periods (log_2_FC = 2, FDR < 0.01).

**Figure 5 ijms-22-01057-f005:**
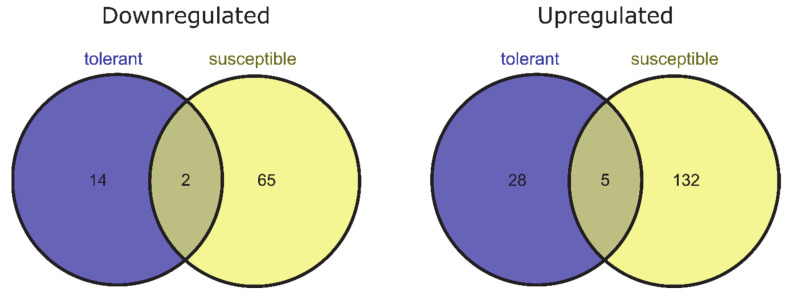
DEGs specific only to de-acclimation during mid-winter warm periods in susceptible and tolerant to de-acclimation barley accessions (log_2_FC = 2, FDR < 0.01).

**Figure 6 ijms-22-01057-f006:**
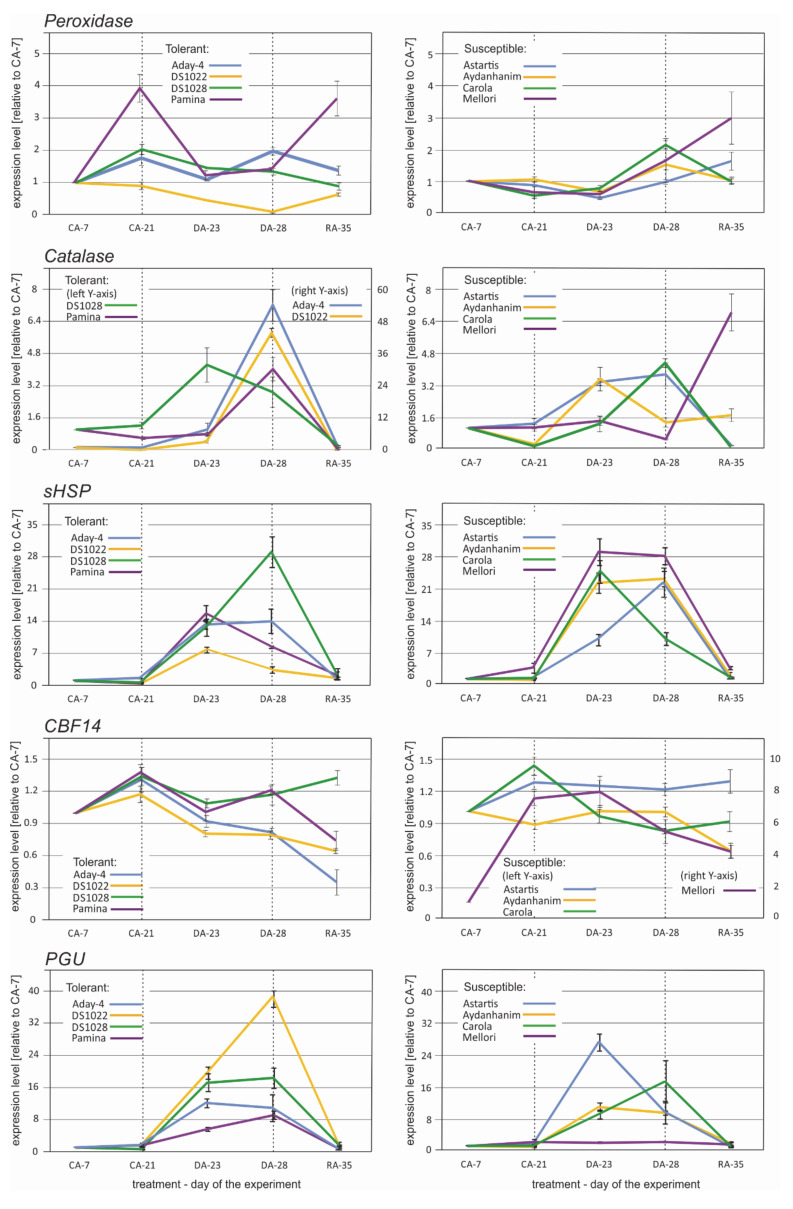
Expression profiles of *peroxidase, catalase, sHSP*, *CBF14, and PGU inhibitor-like* genes during acclimation to cold (CA-7), after 3-week cold acclimation (CA-21), during de-acclimation (DA-23), after 7-day de-acclimation (DA-28), and during re-acclimation to cold (RA-35) in tolerant (left) and susceptible (right) to de-acclimation barley accessions. The de-acclimation period is indicated between the vertical dashed lines.

**Figure 7 ijms-22-01057-f007:**
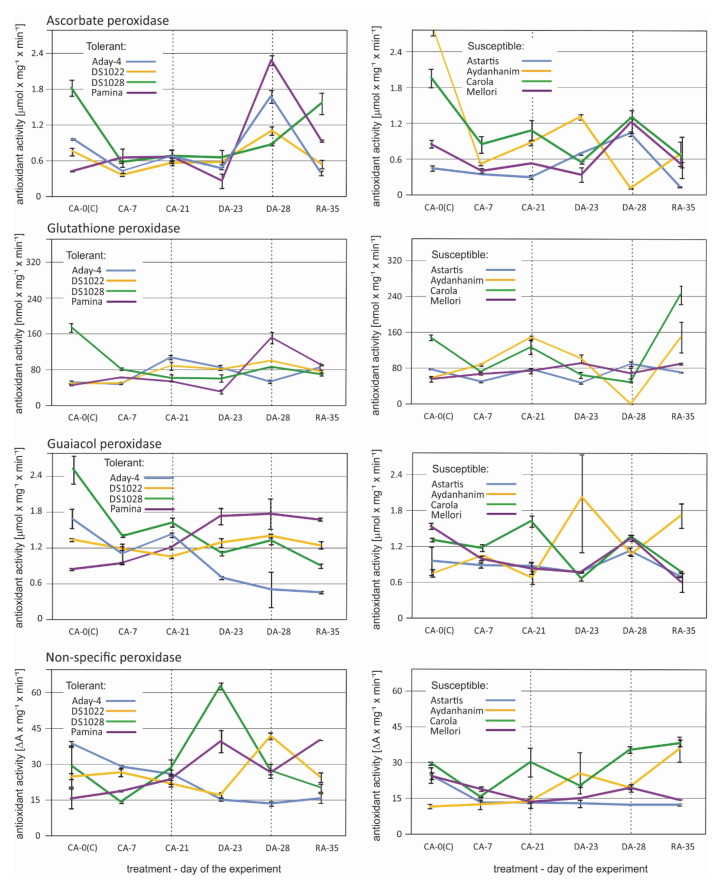
Changes in antioxidant activity of peroxidases: Ascorbate, glutathione, guaiacol, and nonspecific peroxidase in six time points—before cold acclimation (CA-0 (C)), during acclimation to cold (CA-7), after 3-week cold acclimation (CA-21), during de-acclimation (DA-23), after 7-day de-acclimation (DA-28), and during re-acclimation to cold (RA-35) in tolerant (left) and susceptible (right) to de-acclimation barley accessions. The de-acclimation period is indicated between the vertical dashed lines.

**Figure 8 ijms-22-01057-f008:**
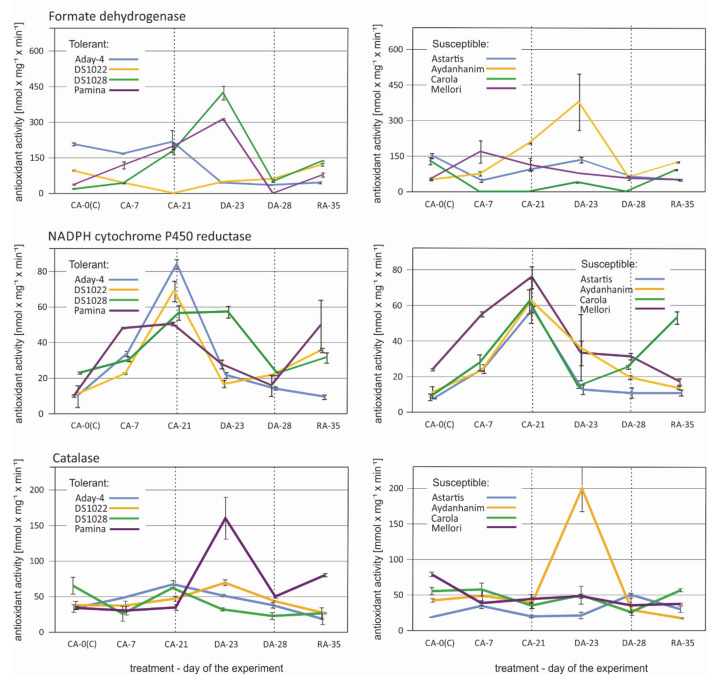
Changes in antioxidant activity of selected enzymes: Formate dehydrogenase, NADPH cytochrome P450 reductase, and catalase in six time points—before cold acclimation (K), during acclimation to cold (CA-0 (C)), during acclimation to cold (CA-7), after 3-week cold acclimation (CA-21), during de-acclimation (DA-23), after 7-day de-acclimation (DA-28), and during re-acclimation to cold (RA-35)in tolerant (left) and susceptible (right) to de-acclimation barley accessions. The de-acclimation period is indicated between the vertical dashed lines.

**Table 1 ijms-22-01057-t001:** Functional groups of significant (FDR < 0.05) GO terms for differentially expressed genes specific only to de-acclimation.

Ontology	Description	Number of DEGs and Direction of Regulation Referred to Cold-Acclimated State; T—Tolerant, S—Susceptible
Biological Process	phosphorylation	39 upregulated (12 T, 27 S)
Biological Process	cellular protein modification process	44 upregulated (13 T, 31 S)
Biological Process	cell recognition	6 upregulated (S)
Biological Process	localization/transport	40 S (25 upregulated, 15 downregulated)
Biological Process	phosphate-containing compound metabolic process	27 upregulated (S)
Biological Process	ion transmembrane transport	8 downregulated (S)
Biological Process	ATP metabolic process	9 downregulated (S)
Biological Process	ribonucleoside monophosphate metabolic process	9 downregulated (S)
Biological Process	cellular nitrogen (including nucleobase-containing) compound metabolic process	24 (23) downregulated (S)
Molecular Function	adenyl ribonucleotide binding	50 upregulated (15 T, 35 S)
Molecular Function	ATP binding	32 upregulated (S)
Molecular Function	transferase activity	44 upregulated (S)
Molecular Function	catalytic activity	91 upregulated (S)

**Table 2 ijms-22-01057-t002:** Annotation of selected differentially expressed barley gene sequences specific only to de-acclimation. Sequences common for both tolerant and susceptible to de-acclimation barley accessions are in italics, N/A—not available.

DEG Number	UniProt Accession Number	Gene	Encoded Protein	BLAST
DEGs downregulated in tolerant to de-acclimation accessions				
HORVU1HR1G085470	A0A287GKC1	N/A	Unknown function protein	26S proteasome non-ATPase regulatory subunit 5 [*Triticum urartu*]
HORVU2HR1G105740	F2DG27	N/A	Predicted protein	late embryogenesis abundant protein 18 [*Aegilops tauschii* subsp. *tauschii*]
HORVU3HR1G084590	A0A287M0T9	N/A	Alkyl transferase	Dehydrodolichyl diphosphate synthase 2 [*Triticum urartu*]
HORVU3HR1G097770	A0A287MG44	N/A	HMA domain-containing protein	heavy metal-associated isoprenylated plant protein 35-like [*Aegilops tauschii* subsp. *tauschii*]
HORVU4HR1G011740	M0VPD2	N/A	Unknown function protein	auxin-repressed 12.5 kDa protein [*Triticum aestivum*]
HORVU5HR1G081720	A0A287S0C8	N/A	Unknown function protein	PREDICTED: *Aegilops tauschii* subsp. *tauschii* pentatricopeptide repeat-containing protein At4g01400, mitochondrial-like (LOC109778148), mRNA
HORVU5HR1G114630	A0A287STL3	N/A	Unknown function protein	pentatricopeptide repeat-containing protein At1g74850, chloroplastic-like [*Aegilops tauschii* subsp. *tauschii*]
HORVU6HR1G037610	A0A287TY77	N/A	Unknown function protein	RNA-binding protein cabeza-like isoform X2 [*Aegilops tauschii* subsp. *tauschii*]
HORVU6HR1G066450	A0A287UH95	N/A	Unknown function protein	MYB-related protein [*Triticum aestivum*]
HORVU6HR1G087460	A0A287V2G5	N/A	Translocase of chloroplast	betaine aldehyde dehydrogenase [*Hordeum vulgare* subsp. *vulgare*]
*HORVU6HR1G091300*	*A0A287V699*	*N/A*	*Unknown function protein*	*P-loop NTPase domain-containing protein LPA1-like* *[Aegilops tauschii subsp. tauschii]*
HORVU7HR1G086180	A0A287X6D2	N/A	Unknown function protein	MscS family inner membrane protein ynaI [*Triticum urartu*]
HORVU7HR1G086810	M0UF38	N/A	Translocase of chloroplast	putative GATA transcription factor 22 [*Aegilops tauschii* subsp. *tauschii*]
*HORVU7HR1G116770*	*F2DRR5*	*N/A*	*Dirigent protein*	*dirigent protein 21-like [Aegilops tauschii subsp. tauschii]*
DEGs upregulated in tolerant to de-acclimation accessions				
*HORVU1HR1G012240*	*M0Y6Q6*	*N/A*	*Unknown function protein*	*adenine/guanine permease AZG1 [Aegilops tauschii subsp. tauschii]*
HORVU1HR1G029850	A0A287F1D6	N/A	Unknown function protein	probable calcium-binding protein CML18 [*Aegilops tauschii* subsp. *tauschii*]
*HORVU1HR1G053440*	*A0A287FMG9*	*N/A*	*Proline dehydrogenase*	*-*
HORVU1HR1G070390	A0A287G4C9	N/A	Exocyst subunit Exo70 family protein	exocyst complex component EXO70B1-like [*Aegilops tauschii* subsp. *tauschii*]
HORVU1HR1G086070	A0A287GKY6	N/A	Auxin-responsive protein	auxin-responsive protein IAA19-like [*Aegilops tauschii* subsp. *tauschii*]
HORVU2HR1G014930	A0A287H5G1	N/A	Unknown function protein	putative lectin receptor-type protein kinase [*Hordeum vulgare* subsp. *vulgare*]
HORVU2HR1G029900	A0A287HJ83	N/A	Unknown function protein	SNF1-type serine-threonine protein kinase [*Triticum polonicum*]
HORVU2HR1G066100	A0A287I9A8	N/A	Unknown function protein	transcription factor bHLH35-like [*Aegilops tauschii* subsp. *tauschii*]
HORVU2HR1G103890	F2DF45	N/A	Predicted protein	putative serine/threonine-protein kinase isoform X1 [*Aegilops tauschii* subsp. *tauschii*]
HORVU2HR1G121310	A0A287JPV5	N/A	Unknown function protein	G-type lectin S-receptor-like serine/threonine-protein kinase At2g19130 [*Setaria italica*]
HORVU3HR1G000150	A0A287JUW2	N/A	Unknown function protein	hypothetical protein TRIUR3_14872 [*Triticum urartu*]
*HORVU3HR1G084170*	*A0A287M0E1*	*N/A*	*Unknown function protein*	*glucan endo-1,3-beta-glucosidase 14-like isoform X1* *[Aegilops tauschii subsp. tauschii]*
HORVU3HR1G084830	A0A287M160	N/A	Unknown function protein	nematode resistance protein-like HSPRO1 [*Aegilops tauschii* subsp. *tauschii*]
HORVU3HR1G086240	A0A287M2U9	N/A	Unknown function protein	elicitor-responsive protein 1-like [*Aegilops tauschii* subsp. *tauschii*]
HORVU3HR1G098150	A0A287MG87	N/A	Unknown function protein	Disease resistance protein RGA2 [*Triticum urartu*]
HORVU4HR1G002710	A0A287MXN9	N/A	N-acetyltransferase domain-containing protein	Acyl-CoA N-acyltransferase (NAT) superfamily protein [*Zea mays*]
HORVU5HR1G034830	M0W8U1	N/A	Unknown function protein	putative WRKY transcription factor 41 [*Triticum urartu*]
HORVU5HR1G042740	M0V1H8	N/A	Glycosyltransferase	crocetin glucosyltransferase, chloroplastic-like [*Aegilops tauschii* subsp. *tauschii*]
*HORVU5HR1G067010*	*A0A287RL26*	*N/A*	*Unknown function protein*	*homeobox-leucine zipper protein HOX11-like* *[Aegilops tauschii subsp. tauschii]*
HORVU5HR1G072020	M0X915	N/A	Unknown function protein	probable WRKY transcription factor 2 [*Aegilops tauschii* subsp. *tauschii*]
*HORVU5HR1G099670*	*A0A287SHF0*	*N/A*	*GAT domain-containing protein*	*target of Myb protein 1-like isoform X1* *[Aegilops tauschii subsp. tauschii]*
HORVU6HR1G012170	A0A287TC05	N/A	Terpene_synth_C domain-containing protein	S-(+)-linalool synthase, chloroplastic-like [*Aegilops tauschii* subsp. *tauschii*]
HORVU6HR1G023340	M0W8I2	N/A	Unknown function protein	Tyrosine-sulfated glycopeptide receptor 1 [*Triticum urartu*]
HORVU6HR1G028220	A0A287TPZ1	N/A	Unknown function protein	Disease resistance protein RPP13 [*Dichanthelium oligosanthes*]
HORVU6HR1G062220	A0A287UE62	N/A	Unknown function protein	phospholipase A1-Ibeta2, chloroplastic-like [*Aegilops tauschii* subsp. *tauschii*]
HORVU7HR1G076310	A0A287WYF9	N/A	Unknown function protein	MADS-box transcription factor 26 isoform X2 [*Aegilops tauschii* subsp. *tauschii*]
HORVU2HR1G094780	A0A287J0H6	N/A	RNase H domain-containing protein	Bidirectional sugar transporter SWEET14 [*Triticum urartu*]
HORVU3HR1G085690	A0A287M208	N/A	Unknown function protein	hypothetical protein TRIUR3_20661 [*Triticum urartu*]
HORVU4HR1G071670	M0VAZ0	N/A	GRAS domain-containing protein	scarecrow-like protein 21 [*Aegilops tauschii* subsp. *tauschii*]
DEGs upregulated in susceptible to de-acclimation accessions				
HORVU0HR1G039970	A0A287EDG6	N/A	Unknown function protein	putative receptor-like protein kinase At3g47110 [*Oryza sativa* Japonica Group]
HORVU1HR1G001950	M0YQK2	N/A	Unknown function protein	12-oxophytodienoic acid reductase 1-A2a [*Triticum aestivum*]
HORVU1HR1G004150	M0ZD08	N/A	Unknown function protein	CI2D [*Hordeum vulgare* subsp. *Vulgare*]
HORVU1HR1G011990	A0A287EN93	N/A	Unknown function protein	CI2D [*Hordeum vulgare* subsp. *Vulgare*]
*HORVU1HR1G012240*	*M0Y6Q6*	*N/A*	*Unknown function protein*	*adenine/guanine permease AZG1* *[Aegilops tauschii subsp. tauschii]*
HORVU1HR1G037250	A0A287F5H3	N/A	ABC transporter domain-containing protein	ABC transporter G family member 28-like isoform X1 [*Aegilops tauschii* subsp. *tauschii*]
HORVU1HR1G040720	A0A287F8T6	N/A	Long-chain-alcohol oxidase	long-chain-alcohol oxidase FAO1-like [*Aegilops tauschii* subsp. *tauschii*]
HORVU1HR1G042370	A0A287FAK6	N/A	Unknown function protein	putative receptor-like protein kinase At4g00960 isoform X1 [*Aegilops tauschii* subsp. *tauschii*]
HORVU1HR1G051450	M0VBP8	N/A	Unknown function protein	urea-proton symporter DUR3 [*Brachypodium distachyon*]
*HORVU1HR1G053440*	*A0A287FMG9*	*N/A*	*Proline dehydrogenase*	*-*
HORVU1HR1G070730	A0A287G4Q8	N/A	Ammonium transporter	ammonium transporter 2 member 1 isoform X2 [*Aegilops tauschii* subsp. *Tauschii*]
HORVU1HR1G072910	A0A287G793	N/A	Unknown function protein	Rop guanine nucleotide exchange factor 1 [*Triticum urartu*]
HORVU1HR1G092240	A0A287GRA5	N/A	Unknown function protein	Glucan endo-1,3-beta-glucosidase 3 [*Triticum urartu*]
HORVU1HR1G093480	A0A287GSJ3	N/A	Unknown function protein	tryptophan synthase alpha subunit [*Secale cereale*]
HORVU2HR1G004280	A0A287GXX3	N/A	Unknown function protein	1-aminocyclopropane-1-carboxylate oxidase homolog 1-like [*Aegilops tauschii* subsp. *tauschii*]
HORVU2HR1G018440	F2CV55	N/A	Peroxidase	Peroxidase 2 [*Triticum urartu*]
HORVU2HR1G018570	F2D6Z5	N/A	Peroxidase	Peroxidase 2 [*Triticum urartu*]
HORVU2HR1G032680	A0A287HLZ8	N/A	Unknown function protein	LRR receptor-like serine/threonine-protein kinase RPK2 [*Aegilops tauschii* subsp. *tauschii*]
HORVU2HR1G038940	F2DNU6	N/A	Predicted protein	photosystem II 10 kDa polypeptide, chloroplastic [*Aegilops tauschii* subsp. *tauschii*]
HORVU2HR1G044520	A0A287HUX4	N/A	Unknown function protein	Cysteine-rich receptor-like protein kinase 25 [*Triticum urartu*]
HORVU2HR1G064160	A0A287I7L8	N/A	Unknown function protein	sugar transport protein 1-like [*Aegilops tauschii* subsp. *tauschii*]
HORVU2HR1G090160	A0A287IW42	N/A	Cytokin-bind domain-containing protein	hypothetical protein BRADI_5g16083v3 [*Brachypodium distachyon*]
HORVU2HR1G094840	A0A287J0D9	N/A	Unknown function protein	RING-H2 finger protein ATL28 [*Triticum urartu*]
HORVU2HR1G098450	A0A287J3Q1	N/A	Unknown function protein	probable LRR receptor-like serine/threonine-protein kinase At1g56130 [*Aegilops tauschii* subsp. *tauschii*]
HORVU2HR1G108180	A0A287JDG1	N/A	Epimerase domain-containing protein	Anthocyanidin reductase [*Triticum urartu*]
HORVU2HR1G116880	A0A287JKB3	N/A	Unknown function protein	Receptor-like serine/threonine-protein kinase SD1-8 [*Triticum urartu*]
HORVU2HR1G117290	A0A287JKU9	N/A	Serine/threonine-protein kinase	G-type lectin S-receptor-like serine/threonine-protein kinase B120 [*Aegilops tauschii* subsp. *tauschii*]
HORVU3HR1G001390	M0YW91	N/A	Unknown function protein	Putative serine/threonine-protein kinase-like protein CCR3 [*Triticum urartu*]
HORVU3HR1G013180	A0A287K4Z4	N/A	Unknown function protein	G-type lectin S-receptor-like serine/threonine-protein kinase At2g19130 [*Aegilops tauschii* subsp. *tauschii*]
HORVU3HR1G013650	A0A287K5J2	N/A	Unknown function protein	glutathione hydrolase 3 [*Oryza sativa* Japonica Group]
HORVU3HR1G019750	M0X3Y1	N/A	Auxin-responsive protein	Auxin-responsive protein IAA16 [*Triticum urartu*]
HORVU3HR1G022780	A0A287KER6	N/A	Unknown function protein	wall-associated receptor kinase-like 20 [*Aegilops tauschii* subsp. *tauschii*]
HORVU3HR1G031020	F2D225	N/A	Predicted protein	cytochrome P450 71A1-like [*Aegilops tauschii* subsp. *tauschii*]
HORVU3HR1G035170	F2DLR2	N/A	Predicted protein	protein TPR1 [*Aegilops tauschii* subsp. *tauschii*]
HORVU3HR1G062170	M0WTL9	N/A	Non-lysosomal glucosylceramidase	non-lysosomal glucosylceramidase-like [*Aegilops tauschii* subsp. *tauschii*]
HORVU3HR1G071470	A0A287LKZ5	N/A	Unknown function protein	ABC transporter G family member 32-like [*Aegilops tauschii* subsp. *tauschii*]
HORVU3HR1G071750	A0A287LLA5	N/A	Unknown function protein	WRKY transcription factor 39 [*Hordeum vulgare* subsp. *vulgare*]
HORVU3HR1G077670	M0Y6H3	N/A	Unknown function protein	disease resistance protein RPS2-like [*Aegilops tauschii* subsp. *tauschii*]
HORVU3HR1G081300	A0A287LX11	N/A	RING-type E3 ubiquitin transferase	U-box domain-containing protein 16 [*Brachypodium distachyon*]
*HORVU3HR1G084170*	*A0A287M0E1*	*N/A*	Unknown function protein	*Glucan endo-1,3-beta-glucosidase 14 [Triticum urartu]*
HORVU3HR1G090170	A0A287M894	N/A	Unknown function protein	tropinone reductase homolog At5g06060-like isoform X3 [*Aegilops tauschii* subsp. *tauschii*]
HORVU3HR1G092460	A0A287MAB7	N/A	Unknown function protein	auxin-responsive protein SAUR71-like [*Aegilops tauschii* subsp. *tauschii*]
HORVU3HR1G096360	A0A287MER1	N/A	Unknown function protein	acyl transferase 4-like [*Aegilops tauschii* subsp. *tauschii*]
HORVU3HR1G109590	M0YBT1	N/A	Unknown function protein	glycosyltransferase [*Triticum aestivum*]
HORVU4HR1G019410	M0ZB44	N/A	Unknown function protein	alkaline invertase [*Triticum aestivum*]
HORVU4HR1G026770	A0A287NMD2	N/A	LEA_2 domain-containing protein	NDR1/HIN1-like protein 13 [*Aegilops tauschii* subsp. *Tauschii*]
HORVU4HR1G052490	A0A287P258	N/A	Unknown function protein	R2R3-MYB protein [*Triticum aestivum*]
HORVU4HR1G071360	F2CZ96	N/A	Cytochrome b561 and DOMON domain-containing protein	cytochrome b561 and DOMON domain-containing protein At4g12980-like [*Aegilops tauschii* subsp. *tauschii*]
HORVU4HR1G072880	A0A287PM62	N/A	Phosphoserine aminotransferase	phosphoserine aminotransferase 1, chloroplastic [*Brachypodium distachyon*]
HORVU4HR1G074250	F2DUL1	N/A	Unknown function protein	BEL1-like homeodomain protein 7 [*Aegilops tauschii* subsp. *tauschii*]
HORVU4HR1G082710	A0A287PVS7	*MLO*	MLO-like protein	MLO [*Triticum aestivum*]
HORVU4HR1G083210	A0A287PXC8	N/A	Unknown function protein	anthranilate phosphoribosyltransferase, chloroplastic [*Aegilops tauschii* subsp. *tauschii*]
HORVU4HR1G084590	F2DV82	N/A	Predicted protein	homeobox protein BEL1 homolog [*Aegilops tauschii* subsp. *tauschii*]
HORVU4HR1G085250	A0A287PY55	N/A	Unknown function protein	tonoplast intrinsic protein [*Hordeum vulgare* subsp. *vulgare*]
HORVU5HR1G000640	A0A287Q624	N/A	Unknown function protein	ATPase 2 [*Hordeum vulgare* subsp. *Vulgare*]
HORVU5HR1G014170	A0A287QEN1	N/A	Unknown function protein	bZIP6 [*Triticum aestivum*]
HORVU5HR1G041660	A0A287QYD2	N/A	Unknown function protein	probable LRR receptor-like protein kinase At1g51890 [*Aegilops tauschii* subsp. *tauschii*]
HORVU5HR1G055850	F2D5N7	N/A	Predicted protein	sugar transport protein 14 [*Aegilops tauschii* subsp. *Tauschii*]
HORVU5HR1G059090	F2DCK3	N/A	Hexosyltransferase	hydroxyproline O-galactosyltransferase GALT3-like [*Aegilops tauschii* subsp. *tauschii*]
HORVU5HR1G061930	A0A287RGQ6	N/A	Unknown function protein	cytochrome P450 71A1-like [*Aegilops tauschii* subsp. *tauschii*]
HORVU5HR1G061950	A0A287RH10	N/A	Unknown function protein	cytochrome P450 71A1-like [*Aegilops tauschii* subsp. *tauschii*]
HORVU5HR1G064020	F2DY24	N/A	Predicted protein	hypothetical protein TRIUR3_10720 [*Triticum urartu*]
HORVU5HR1G066360	M0ZDV2	N/A	Unknown function protein	putative high-affinity sulfate transporter [*Triticum turgidum* subsp. *durum*]
*HORVU5HR1G067010*	*A0A287RL26*	*N/A*	*Unknown function protein*	*homeobox-leucine zipper protein HOX11 [Brachypodium distachyon]*
HORVU5HR1G070360	A0A287RPR9	N/A	Unknown function protein	putative wall-associated receptor kinase-like 16 [*Aegilops tauschii* subsp. *tauschii*]
HORVU5HR1G080340	M0YYJ0	*CBF12C*	CBF12C	CBF12C [*Hordeum vulgare* subsp. *Vulgare*]
HORVU5HR1G080350	Q3SAT4	*CBF14*	CBF14	HvCBF14 [*Hordeum vulgare* subsp. *Vulgare*]
HORVU5HR1G080860	A0A287RZG2	N/A	Unknown function protein	UDP-D-glucose epimerase 3 [*Hordeum vulgare*]
HORVU5HR1G085710	D2KZ48	*HvNIP1;2*	Nodulin-26 like intrinsic protein	nodulin-26 like intrinsic protein [*Hordeum vulgare* subsp. *vulgare*]
HORVU5HR1G093090	A0A287S9R4	N/A	AA_permease_C domain-containing protein	cationic amino acid transporter 1-like [*Aegilops tauschii* subsp. *tauschii*]
HORVU5HR1G093660	F2EFG2	N/A	Nonspecific serine/threonine protein kinase	CBL-interacting protein kinase 7 [*Triticum aestivum*]
HORVU5HR1G094430	A0A287SBM8	N/A	Unknown function protein	cadmium/zinc-transporting P1B-ATPase 3 isoform HMA3.1 [*Hordeum vulgare* subsp. *vulgare*]
HORVU5HR1G097270	A0A287SF79	N/A	Peroxidase	peroxidase 50-like [*Aegilops tauschii* subsp. *tauschii*]
*HORVU5HR1G099670*	*A0A287SHF0*	*N/A*	*GAT domain-containing protein*	*target of Myb protein 1-like isoform X1* *[Aegilops tauschii subsp. tauschii]*
HORVU5HR1G105930	M0UPG3	N/A	CASP-like protein	CASP-like protein 1U3 [*Aegilops tauschii* subsp. *tauschii*]
HORVU5HR1G109040	A0A287SNK5	N/A	Unknown function protein	lipid transfer protein [*Triticum aestivum*]
HORVU5HR1G110180	A0A287SPK7	N/A	Unknown function protein	phosphate transporter [*Hordeum vulgare* subsp. *vulgare*]
HORVU6HR1G008640	A0A287T8X2	N/A	Catalase	RecName: Full = Catalase isozyme 2 [*Hordeum vulgare*]
HORVU6HR1G037850	F2DCG6	N/A	Predicted protein	F-box/kelch-repeat protein At1g55270-like [*Aegilops tauschii* subsp. *tauschii*]
HORVU6HR1G061280	A0A287UDJ5	N/A	Unknown function protein	probable receptor-like protein kinase At1g33260 [*Aegilops tauschii* subsp. *tauschii*]
HORVU6HR1G061450	M0XJI0	N/A	Unknown function protein	iron-phytosiderophore transporter [*Hordeum vulgare* subsp. *Vulgare*]
HORVU6HR1G069400	A0A287UKB1	N/A	2-Hacid_dh domain-containing protein	formate dehydrogenase [*Triticum aestivum*]
HORVU6HR1G073660	M0VY04	N/A	Unknown function protein	nudix hydrolase 17, mitochondrial-like [*Aegilops tauschii* subsp. *tauschii*]
HORVU6HR1G076510	F2DC11	N/A	Predicted protein	metalloendoproteinase 1 precursor [*Zea mays*]
HORVU7HR1G007480	A0A287VFS1	N/A	Unknown function protein	probable LRR receptor-like serine/threonine-protein kinase At3g47570 [*Brachypodium distachyon*]
HORVU7HR1G007520	A0A287VFT7	N/A	Unknown function protein	LRR receptor-like serine/threonine-protein kinase FLS2 [*Triticum urartu*]
HORVU7HR1G011250	A0A287VI73	N/A	Unknown function protein	probable E3 ubiquitin-protein ligase XBOS36 [*Aegilops tauschii* subsp. *tauschii*]
HORVU7HR1G013710	A0A287VJQ7	N/A	3-phosphoshikimate 1-carboxyvinyltransferase	chloroplast 5-enolpyruvylshikimate-3-phosphate synthase [*Triticum aestivum*]
HORVU7HR1G024670	F2D5K9	N/A	Predicted protein	UCW116, putative lipase [*Hordeum vulgare* subsp. *Vulgare*]
HORVU7HR1G025670	A0A287VRC7	N/A	Unknown function protein	protein NRT1/PTR FAMILY 2.3-like [*Aegilops tauschii* subsp. *tauschii*]
HORVU7HR1G030380	A0A287VWN6	N/A	Unknown function protein	cinnamoyl-CoA reductase 1-like [*Aegilops tauschii* subsp. *tauschii*]
HORVU7HR1G048970	F2D5L5	N/A	Predicted protein	tricin synthase 1-like [*Aegilops tauschii* subsp. *tauschii*]
HORVU7HR1G052560	A0A287WI67	N/A	Calcium-transporting ATPase	calcium-transporting ATPase 8, plasma membrane-type-like isoform X1 [*Aegilops tauschii* subsp. *tauschii*]
HORVU7HR1G086580	A0A287X6Z2	N/A	Unknown function protein	U-box domain-containing protein 34-like [*Aegilops tauschii* subsp. *tauschii*]
HORVU7HR1G089360	A0A287X975	N/A	Peroxidase	peroxidase P7-like [*Aegilops tauschii* subsp. *tauschii*]
HORVU7HR1G108530	A0A287XRG5	N/A	Peroxidase	peroxidase 70-like [*Aegilops tauschii* subsp. *tauschii*]
HORVU7HR1G113510	A0A287XV75	N/A	Unknown function protein	L-type lectin-domain-containing receptor kinase IX.1-like [*Aegilops tauschii* subsp. *tauschii*]
HORVU7HR1G114660	A0A287XVT0	N/A	Unknown function protein	Indole-3-glycerol phosphate synthase, chloroplastic [*Triticum urartu*]
HORVU1HR1G002410	A0A287EGE7	N/A	LRRNT_2 domain-containing protein	polygalacturonase inhibitor-like [*Aegilops tauschii* subsp. *tauschii*]
HORVU1HR1G011430	M0UHV3	N/A	IU_nuc_hydro domain-containing protein	unnamed protein product [*Triticum turgidum* subsp. *durum*]
HORVU1HR1G011730	A0A287EMY8	N/A	Protein kinase domain-containing protein	receptor like protein kinase S.2-like isoform X1 [*Aegilops tauschii* subsp. *tauschii*]
HORVU1HR1G023220	A0A287EX35	N/A	Unknown function protein	predicted protein [*Hordeum vulgare* subsp. *vulgare*]
HORVU1HR1G052010	A0A287FKD1	N/A	C2 NT-type domain-containing protein	protein PLASTID MOVEMENT IMPAIRED 1-like [*Aegilops tauschii* subsp. *tauschii*]
HORVU1HR1G070720	A0A287G4Q3	N/A	Unknown function protein	unknown [*Zea mays*]
HORVU1HR1G080860	A0A287GGB7	N/A	Unknown function protein	predicted protein [*Hordeum vulgare* subsp. *vulgare*]
HORVU2HR1G010560	A0A287H1T4	N/A	Unknown function protein	-
HORVU2HR1G019180	A0A287H8Z0	N/A	NAD(P)-bd_dom domain-containing protein	UDP-D-glucuronate decarboxylase [*Hordeum vulgare*]
HORVU2HR1G030660	M0VWF8	N/A	Unknown function protein	unnamed protein product [*Triticum turgidum* subsp. *durum*]
HORVU2HR1G032360	A0A287HLF8	N/A	Unknown function protein	predicted protein [*Hordeum vulgare* subsp. *vulgare*]
HORVU2HR1G096230	M0UYR4	N/A	Aa_trans domain-containing protein	lysine histidine transporter-like 8 [*Brachypodium distachyon*]
HORVU3HR1G027700	A0A287KJ28	N/A	DJ-1_PfpI domain-containing protein	protein DJ-1 homolog B-like [*Aegilops tauschii* subsp. *tauschii*]
HORVU3HR1G073280	M0WFD0	N/A	Unknown function protein	hypothetical protein TRIUR3_04936 [*Triticum urartu*]
HORVU3HR1G075150	F2D9C7	N/A	Predicted protein	protein LURP-one-related 5-like [*Aegilops tauschii* subsp. *tauschii*]
HORVU4HR1G009140	A0A287N4V6	N/A	PfkB domain-containing protein	putative ribokinase [*Triticum turgidum* subsp. *Durum*]
HORVU4HR1G060260	M0Y821	N/A	Unknown function protein	predicted protein [*Hordeum vulgare* subsp. *vulgare*]
HORVU4HR1G067840	M0YH93	N/A	TPT domain-containing protein	GDP-mannose transporter GONST3-like isoform X1 [*Aegilops tauschii* subsp. *tauschii*]
HORVU4HR1G076750	M0WWI7	N/A	AB hydrolase-1 domain-containing protein	salicylic acid-binding protein 2-like [*Aegilops tauschii* subsp. *tauschii*]
HORVU4HR1G085150	M0UZW0	N/A	Unknown function protein	unnamed protein product [*Triticum turgidum* subsp. *durum*]
HORVU5HR1G002340	A0A287Q7I1	N/A	Unknown function protein	predicted protein [*Hordeum vulgare* subsp. *vulgare*]
HORVU5HR1G040970	A0A287QXS7	N/A	DUF3700 domain-containing protein	stem-specific protein TSJT1 [*Aegilops tauschii* subsp. *tauschii*]
HORVU5HR1G049370	A0A287R4Y8	N/A	Hydrolase_4 domain-containing protein	caffeoylshikimate esterase [*Brachypodium distachyon*]
HORVU5HR1G084700	F2EJL0	N/A	Predicted protein	predicted protein [*Hordeum vulgare* subsp. *vulgare*]
HORVU5HR1G104050	A0A287SK30	N/A	Unknown function protein	purine permease 3-like [*Aegilops tauschii* subsp. *tauschii*]
HORVU6HR1G070610	A0A287UL09	N/A	Unknown function protein	M55 family metallopeptidase [*Oscillibacter* sp. 1–3]
HORVU6HR1G093860	M0YBQ9	N/A	Unknown function protein	protein LAZY 1 [*Aegilops tauschii* subsp. *tauschii*]
HORVU7HR1G036780	A0A287W267	N/A	Unknown function protein	protein EXORDIUM-like [*Aegilops tauschii* subsp. *tauschii*]
DEGs downregulated in susceptible to de-acclimation accessions				
HORVU0HR1G002870	A0A287DV08	N/A	Unknown function protein	E3 ubiquitin-protein ligase RDUF1-like [*Aegilops tauschii* subsp. *tauschii*]
HORVU0HR1G003900	A0A287DVH9	N/A	Unknown function protein	glycine-rich cell wall structural protein-like [*Aegilops tauschii* subsp. *tauschii*]
HORVU0HR1G021760	M0V2B7	N/A	Unknown function protein	ent-kaurene oxidase 1 [*Hordeum vulgare* subsp. *Vulgare*]
HORVU1HR1G052560	A0A287FKY6	N/A	Unknown function protein	DDB1- and CUL4-associated factor 8 [*Triticum urartu*]
HORVU1HR1G053080	A0A287FLT1	N/A	Unknown function protein	carotene epsilon-monooxygenase, chloroplastic [*Aegilops tauschii* subsp. *tauschii*]
HORVU1HR1G076190	A0A287GAV2	N/A	Unknown function protein	heat shock protein 101 [*Triticum aestivum*]
HORVU1HR1G083420	F2D3K2	*HsfA2c*	Heat shock factor A2c	heat shock factor A2c [*Hordeum vulgare* subsp. *vulgare*]
HORVU1HR1G087070	A0A287GLN9	N/A	Unknown function protein	DnaJ homolog subfamily B member 13 [*Triticum urartu*]
HORVU1HR1G094480	A0A287GTJ4	N/A	ATP-dependent Clp protease proteolytic subunit	PREDICTED: ATP-dependent Clp protease proteolytic subunit-related protein 1, chloroplastic [*Oryza brachyantha*]
HORVU2HR1G071860	A0A287IEE1	N/A	Predicted protein	3-beta hydroxysteroid dehydrogenase/isomerase family protein [*Zea mays*]
HORVU2HR1G076530	M0VD73	N/A	Unknown function protein	hypothetical protein TRIUR3_05260 [*Triticum urartu*]
HORVU2HR1G081670	M0WB36	N/A	Unknown function protein	ATP-dependent 6-phosphofructokinase 2 [*Brachypodium distachyon*]
HORVU3HR1G030950	F2DNE0	N/A	Predicted protein	cytochrome P450 71A1-like [*Aegilops tauschii* subsp. *tauschii*]
HORVU3HR1G042770	A0A287KWA3	N/A	Predicted protein	putative anion transporter 1, chloroplastic [*Triticum urartu*]
HORVU3HR1G063620	A0A287LBU9	N/A	Unknown function protein	65-kDa microtubule-associated protein 3 [*Brachypodium distachyon*]
HORVU3HR1G067380	A0A287LGC6	N/A	Unknown function protein	probable protein phosphatase 2C 50 [*Aegilops tauschii* subsp. *tauschii*]
HORVU3HR1G074780	A0A287LPG5	N/A	Unknown function protein	protein STRICTOSIDINE SYNTHASE-LIKE 10-like [*Aegilops tauschii* subsp. *tauschii*]
HORVU3HR1G078270	A0A287LT80	N/A	Unknown function protein	type II metacaspase [*Triticum aestivum*]
HORVU3HR1G081960	A0A287LXY1	N/A	SUEL-type lectin domain-containing protein	beta-galactosidase 3-like [*Aegilops tauschii* subsp. *tauschii*]
HORVU3HR1G085100	A0A287M173	N/A	Unknown function protein	protein ENHANCED PSEUDOMONAS SUSCEPTIBILTY 1 [*Brachypodium distachyon*]
HORVU3HR1G089300	A0A287M6P9	N/A	Unknown function protein	dehydrin [*Hordeum vulgare* subsp. *Vulgare*]
HORVU3HR1G106720	A0A287MKT2	N/A	Pyrroline-5-carboxylate reductase	pyrroline-5-carboxylate reductase [*Triticum aestivum*]
HORVU3HR1G113340	A0A287MR98	N/A	Unknown function protein	probable aspartyl aminopeptidase [*Aegilops tauschii* subsp. *tauschii*]
HORVU4HR1G018180	A0A287NFC0	N/A	Unknown function protein	methylsterol monooxygenase 1–2 [*Brachypodium distachyon*]
HORVU4HR1G027150	M0XWK0	N/A	Unknown function protein	Mitochondrial uncoupling protein 3 [*Triticum urartu*]
HORVU4HR1G072620	M0XE93	N/A	Unknown function protein	ABC transporter G family member 22 [*Triticum urartu*]
HORVU4HR1G085590	F2DTJ0	N/A	Predicted protein	subtilisin-like protease SBT1.2 [*Brachypodium distachyon*]
HORVU5HR1G006780	A0A287Q9A7	N/A	Unknown function protein	5-methyltetrahydropteroyltriglutamate—homocysteine methyltransferase 1-like [*Aegilops tauschii* subsp. *tauschii*]
HORVU5HR1G056620	A0A287RBF6	N/A	Unknown function protein	NAD kinase 4 [*Triticum aestivum*]
HORVU5HR1G059860	A0A287RE25	N/A	Unknown function protein	DEAD-box ATP-dependent RNA helicase 22 [*Triticum urartu*]
HORVU5HR1G062310	F2DDU3	N/A	Predicted protein	20 kDa chaperonin, chloroplastic-like [*Aegilops tauschii* subsp. *tauschii*]
HORVU5HR1G106790	A0A287SML9	N/A	Unknown function protein	probable GTP-binding protein OBGC2 isoform X1 [*Oryza sativa* Japonica Group]
HORVU6HR1G031480	A0A287TST0	N/A	Succinate-semialdehyde dehydrogenase	succinate-semialdehyde dehydrogenase, mitochondrial [*Aegilops tauschii* subsp. *Tauschii*]
HORVU6HR1G065710	A0A287UGU3	N/A	Unknown function protein	SPX domain-containing membrane protein Os02g45520 [*Aegilops tauschii* subsp. *tauschii*]
HORVU6HR1G068490	M0WYY1	N/A	LAGLIDADG_2 domain-containing protein	pentatricopeptide repeat-containing protein OTP51, chloroplastic [*Brachypodium distachyon*]
HORVU6HR1G069260	A0A287UK48	N/A	Unknown function protein	universal stress protein A-like protein [*Aegilops tauschii* subsp. *tauschii*]
HORVU6HR1G077710	A0A287UT21	N/A	SHSP domain-containing protein	24.1 kDa heat shock protein, mitochondrial-like isoform X2 [*Aegilops tauschii* subsp. *tauschii*]
*HORVU6HR1G091300*	*A0A287V699*	*N/A*	*Unknown function protein*	*P-loop NTPase domain-containing protein LPA1-like* *[Aegilops tauschii subsp. tauschii]*
HORVU6HR1G095270	A0A287VAF7	N/A	Protein DETOXIFICATION	protein DETOXIFICATION 45, chloroplastic isoform X1 [*Aegilops tauschii* subsp. *tauschii*]
HORVU7HR1G027560	M0WGG7	N/A	Unknown function protein	CONSTANS-like protein CO8 [*Hordeum vulgare* subsp. *vulgare*]
HORVU7HR1G034990	A0A287W0Z9	N/A	Kinesin-like protein	kinesin-like protein KIN-7D, chloroplastic isoform X1 [*Aegilops tauschii* subsp. *tauschii*]
HORVU7HR1G047700	F2CWR0	N/A	Formate dehydrogenase, mitochondrial	RecName: Full = Formate dehydrogenase, mitochondrial; Short = FDH; AltName: Full = NAD-dependent formate dehydrogenase; Flags: Precursor [*Hordeum vulgare*]
HORVU7HR1G098580	A0A287XIG8	N/A	Lipase_GDSL domain-containing protein	acetylajmalan esterase-like [*Aegilops tauschii* subsp. *tauschii*]
HORVU7HR1G110170	A0A287XSX0	N/A	Histone H2B	Histone H2B.2 [*Triticum urartu*]
*HORVU7HR1G116770*	*F2DRR5*	*N/A*	*Dirigent protein*	*dirigent protein 21-like [Aegilops tauschii subsp. tauschii]*
HORVU0HR1G034940	A0A287KZ68	N/A	Unknown function protein	putative protein 137 [*Sorghum arundinaceum*]
HORVU1HR1G030000	A0A287F1D9	N/A	Unknown function protein	predicted protein [*Hordeum vulgare* subsp. *vulgare*]
HORVU1HR1G072220	A0A287G6E5	N/A	Unknown function protein	Heat shock cognate 70 kDa protein 4 [*Triticum urartu*]
HORVU1HR1G078350	M0Y7A7	N/A	Methyltransf_11 domain-containing protein	phosphomethylethanolamine N-methyltransferase-like [*Aegilops tauschii* subsp. *tauschii*]
HORVU2HR1G087490	A0A287ITE4	N/A	Unknown function protein	protein trichome birefringence-like 10 [*Aegilops tauschii* subsp. *tauschii*]
HORVU3HR1G048810	A0A287KZ68	N/A	Unknown function protein	putative protein 137 [*Sorghum arundinaceum*]
HORVU3HR1G059610	A0A287L8L9	N/A	Unknown function protein	Phosphoglycerate mutase family protein [*Zea mays*]
HORVU4HR1G065180	F2DC44	N/A	Predicted protein	predicted protein [*Hordeum vulgare* subsp. *vulgare*]
HORVU5HR1G016810	A0A287QGT4	N/A	HVA22-like protein	HVA22-like protein e [*Aegilops tauschii* subsp. *tauschii*]
HORVU5HR1G069360	F2DJC5	N/A	Predicted protein	oil body-associated protein 1A [*Brachypodium distachyon*]
HORVU6HR1G049490	A0A287KZ68	N/A	Unknown function protein	putative protein 137 [*Sorghum arundinaceum*]
HORVU6HR1G060020	M0Z5W3	N/A	BRO1 domain-containing protein	hypothetical protein TRIUR3_13310 [*Triticum urartu*]
HORVU6HR1G063480	A0A287UES2	N/A	PALP domain-containing protein	cysteine synthase-like isoform X1 [*Aegilops tauschii* subsp. *tauschii*]
HORVU6HR1G073040	A0A287UNF9	N/A	MADS-box domain-containing protein	MADS-box transcription factor TaAGL17 [*Triticum aestivum*]
HORVU6HR1G082360	A0A287UXN4	N/A	SHSP domain-containing protein	18.6 kDa class III heat shock protein-like [*Aegilops tauschii* subsp. *tauschii*]
HORVU7HR1G051730	A0A287WH78	N/A	Unknown function protein	unnamed protein product [*Triticum turgidum* subsp. *durum*]
HORVU7HR1G091670	A0A287KZ68	N/A	Unknown function protein	putative protein 137 [*Sorghum arundinaceum*]

**Table 3 ijms-22-01057-t003:** Primer and probe sequences in the expression analysis of selected candidate genes associated with tolerance to de-acclimation in winter barley.

Gene	Forward Primer 3′-5′	Reverse Primer 3′-5′	Probe 3′-5′
Peroxidase	GCACTTCCACGACTGCTTTG	CCATGCCAGACAGCAGAACA	FAM-CCAAGGTTGTGACGCGT-MGB
Catalase	GGACCTGCTCGGCAACAA	GGGCTTGAAGGCGTGGAT	FAM-CCCCGTCTTCTTCA-MGB
CBF14	CAGCATCCATCTCTCCCAAGTC	TGTGGAGTAAGCAGCGTGTTTT	FAM-CAGCGCAGCAGCT-MGB
PGU inhibitor-like	TACCACTTTGCGTCCTGGAC	TCAGCATCACAGTCGACGTC	FAM-GCCCGACTCCGCCTGTTGC-MGB
sHSP	GTCGCCATCGCCTGATCT	TGACAAACGCCGATGAGGTA	FAM-TACCTCAGTCGCGCCAG-MGB

## Data Availability

The data supporting the findings of this study are available from the corresponding author upon reasonable request.

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
