# Peer review of "Identification of the Genetic Basis of Response to de-Acclimation in Winter Barley"

_ijms, 2021, doi:10.3390/ijms22031057_

Round 1
Reviewer 1 Report
This manuscript gives novel insights into the complex processes that take place in barley during deacclimatization. The manuscript is very well written. Methodology is clearly explained. The authors have found quite an impressive number of differentially expressed genes under the experimental conditions and analyzed some of them further. The results show that in the future tolerance of barley to deacclimation or its ability for rapid reacclimation are crucial for winter hardiness in the future. Differences between the data shown in this paper, obtained on a monocot (barley) and literature data referring to dicots (Arabidopsis) are clearly shown. The results of the research presented in this manuscript may be very useful for future barley breeding but also for modeling of the effects of expected climate changes on winter crops. Additional proteomic analyses and gene annotation in the future are expected to decipher these compound processes further.
Author Response
We thank the Reviewer for taking time to review our manuscript and for his positive opinion.
Reviewer 2 Report
- Very good work.
- Please enrich the abstract (at the end) with a part of your final conclusions (Line 458-474). You have very interesting conclusions but you make a small reference in the abstract.
- A part of the introduction should be written again. “Lines 39-42. Under global warming, it might be considered that winter hardiness will be less critical for future crop production. However, this assumption is invalid, as the only parameters likely to change will be the predominant factors that influence overwintering of plants locally.” . I propose to author to write in the intoduction that climate change senarios cause unstable wheather condition which many time are ubnormal to the usual season weather condition. Something like that will help the authrors to connect clima scenarios with winter hardiness.
- In the introduction, in order to support your research, please and more references in lines 73-78, “Winter barley shows a relatively weak cold acclimation capability [8] and, in consequence, low winter hardiness, which limits large-scale production of the crop despite increasing interest from the beer industry in winter barley cultivars. The genetic basis of freezing tolerance in winter barley has been studied previously by many research groups, for example [9]” .
- At the end of the introduction please write clearly the aim of the study.
- English language and grammar control it is necessary to be improved. I propose many changes in the pdf, but many other mistakes exist.

Author Response
- Please enrich the abstract (at the end) with a part of your final conclusions (Line 458-474). You have very interesting conclusions but you make a small reference in the abstract.
We have added two sentences based on the final conclusions as the Reviewer suggested.
- A part of the introduction should be written again. “Lines 39-42. Under global warming, it might be considered that winter hardiness will be less critical for future crop production. However, this assumption is invalid, as the only parameters likely to change will be the predominant factors that influence overwintering of plants locally.” . I propose to author to write in the intoduction that climate change senarios cause unstable wheather condition which many time are ubnormal to the usual season weather condition. Something like that will help the authrors to connect clima scenarios with winter hardiness.
We modified the Introduction by adding the information about future climate scenarios, as suggested by the Reviewer.
- In the introduction, in order to support your research, please and more references in lines 73-78, “Winter barley shows a relatively weak cold acclimation capability [8] and, in consequence, low winter hardiness, which limits large-scale production of the crop despite increasing interest from the beer industry in winter barley cultivars. The genetic basis of freezing tolerance in winter barley has been studied previously by many research groups, for example [9]” .
We added five references to this section.
- At the end of the introduction please write clearly the aim of the study.
We added a new paragraph at the end of Introduction. It describes the aim of the study as well as the research hypothesis.
- English language and grammar control it is necessary to be improved. I propose many changes in the pdf, but many other mistakes exist.
The manuscript was subjected to English editing prior to the original submission and was edited thoroughly by Robert McKenzie, PhD, from Edanz Group (https://en-author-services.edanz.com/ac), who we thank in the acknowledgements section. We also attach Edanz Editing Certificate which confirms that the editing took place. Nevertheless we are thankful for the Reviewer’s suggestions and after consultation with a specialist we implemented some of the proposed changes.

Round 2
Reviewer 2 Report
I am glad to notice that the majority of the proposed changes have been incorporated in the manuscript improving it and make it proper for publication.